# Early induction of cytokine release syndrome by rapidly generated CAR T cells in preclinical models

Arezoo Jamali [ID][1,5], Naphang Ho [ID][1,2,5], Angela Braun[1,3], Elham Adabi[1], Frederic B Thalheimer [ID][1,2,4] & Christian J Buchholz [ID][1,2,3 ✉]

## Abstract

**Cytokine release syndrome (CRS) is a significant side-effect of conventional chimeric antigen receptor (CAR) T-cell therapy. To facilitate patient accessibility, short-term (st) CAR T cells, which are administered to patients only 24 h after vector exposure, are in focus of current investigations. Their impact on the incidence and severity of CRS has been poorly explored. Here, we evaluated CD19-specific stCAR T cells in preclinical models. In co-culture with tumor cells and monocytes, stCAR T cells exhibited anti-tumoral activity and potent release of CRS-related cytokines (IL-6, IFN-γ, TNF-α, GM-CSF, IL-2, IL-10). When administered to NSG-SGM3 mice, stCAR T cells, but not conventional CAR T cells, induced severe acute adverse events within 24 h, including hypothermia and weight loss, as well as high body scores, independent of the presence of tumor target cells. Human (IFN-γ, TNF-α, IL-2, IL-10) and murine (MCP-1, IL-6, G-CSF) cytokines, typical for severe CRS, were systemically elevated. Our data highlight potential safety risks of rapidly manufactured CAR T cells and suggest NSG-SGM3 mice as sensitive model for their preclinical safety evaluation.**

**Keywords** NSG-SGM3 Mouse Model; Myeloid Cells; Rapid Manufactured CAR T Cell; CRS Model; lentiviral Vector
**Subject Categories** Cancer; Immunology

## Introduction

CAR T cells have vibrantly transformed the field of cancer immunotherapy with rapid progress as a new therapeutic modality with promising outcomes in hematologic malignancies. The extensive efforts over the last two decades have resulted in outstanding achievements with several CAR T-cell products having reached the market. However, a major drawback of current CAR T-cell products is the long manufacturing time, which can last weeks before the patient receives the therapeutic cell product. (Levine et al, 2017; Vormittag et al, 2018; Abou-El-Enein et al, 2021).

Therefore, the generation of CAR T cells within a few days is currently in research focus. The benefits of such CAR T cells would not only be an accelerated availability for patients but potentially also a more advantageous cell composition of the CAR T-cell product. For instance, CAR T cells generated within 3 days display a more naive-like phenotype and exhibit superior proliferation and effector function post adoptive cell transfer into tumor-engrafted mice, in contrast to long-term expanded CAR T cells. (Ghassemi et al, 2018). While current studies reveal that short-term (st) CAR T cells may provide a beneficial phenotype outperforming conventional CAR T cells by in vivo expansion and anti-tumor activity (Ghassemi et al, 2018; Zhang et al, 2022; Yang et al, 2022), the safety of these cells is less well characterized. Especially, the short time period after exposure to vector particles prior to administration harbors the likelihood of incomplete transduction, making the properties of stCAR T cells rather unpredictable. So far, preclinical studies and mouse models used were selected to focus specifically on the therapeutic activities of stCAR T cells. (Ghassemi et al, 2018; Zhang et al, 2022; Yang et al, 2022; Ghassemi et al, 2022). Assay systems identifying residual vector particles as well as a careful safety assessment of these CAR T cells are therefore urgently needed.

Cytokine release syndrome (CRS) and immune effector cell-associated neurotoxicity syndrome (ICANS) represent the most prominent side effects caused by conventional CAR T cells, which can reach fatal states in particular patients (Morris et al, 2022). Clinically, CRS is characterized by fever at the onset, followed by hypotension, capillary leak, and organ dysfunction with critical medical care requirements (Lee et al, 2019). The severity of CRS correlates to tumor burden, the amount of infused CAR T cells and their proliferation (Morris et al, 2022; Donnadieu et al, 2022). Commonly, CRS appears a few days after CAR T-cell administration and peaks within a week. During this period, high CAR T-cell expansion can be observed, which is associated with massive elevation of various inflammatory cytokines in the serum, including IL-6, IL-10, IL-8, IFN-γ, and TNF-α (Morris et al, 2022; Lee et al, 2019; Donnadieu et al, 2022). Preclinical studies in mouse models showed that CRS is a complex interplay between activated CAR T cells encountering tumor antigens and further immune cells (Giavridis et al, 2018; Norelli et al, 2018). In particular, monocytes and macrophages have been identified as a major source of IL-6

[1]Molecular Biotechnology and Gene Therapy, Paul-Ehrlich-Institut, Langen, Germany. [2]Frankfurt Cancer Institute, Goethe University, Frankfurt am Main, Germany. [3]German Cancer Consortium (DKTK) and German Cancer Research Center (DKFZ), Heidelberg, Germany. [4]Hematology, Cell and Gene Therapy (HZG), Paul-Ehrlich-Institut, Langen, Germany. [5]These authors contributed equally: Arezoo Jamali, Naphang Ho. ✉E-mail: Christian.Buchholz@pei.de

and IL-1 and can therefore be considered as key drivers for induction and progression of CRS. (Giavridis et al, 2018; Norelli et al, 2018; Barrett et al, 2016; Singh et al, 2017; Sachdeva et al, 2019). Although CRS can be managed by cytokine receptor blockers, e.g., tocilizumab, it remains in focus of ongoing research, including the identification of predictive preclinical model systems (Giavridis et al, 2018; Norelli et al, 2018; Singh et al, 2017; Sachdeva et al, 2019; Arcangeli et al, 2022).

Here, we focused on the preclinical safety evaluation of stCAR T cells, which were generated within 3 days after T-cell isolation. We show that a high percentage of stCAR T cells have viral vector particles attached to their surface, which disappear over cultivation time. Their cytotoxic activity was confirmed in an in vitro killing assay, with enhanced secretion of CRS-relevant cytokines in the presence of monocytes. When administered to NSG-SGM3 mice, stCAR T cells, but not conventional CAR T cells, rapidly induced severe symptoms, and cytokine release independently of the presence of tumor target cells. This was confirmed in the ex vivo assay, where co-cultivation of stCAR T cells and monocytes was sufficient to release CRS-relevant cytokines.

# Results

## In vitro characterization of short-term CAR T cells

To generate CD19-specific stCAR T cells, human PBMCs were activated for 2 days with anti-CD3 and anti-CD28 antibodies in the presence of IL-7 and IL-15 and subsequently incubated with VSV-LV (MOI 4–5) for 24 h (Fig. 1A). Then, stCAR T cells were analyzed by flow cytometry to detect the VSV glycoprotein (VSV-G) and CAR expression via myc tag. About 62% of the cells were single-positive for VSV-G, while a smaller proportion of about 24% were additionally positive for CAR (Fig. 1B,C). Less than 2% of the cells were solely positive for the CAR with no detectable VSV-G. For conventional CAR T-cell production, human PBMCs were activated in the same way, but LVs were added on day 3 post-activation followed by another 3 days of cultivation. These CAR T cells contained substantially lower amounts of detectable VSV-G protein (Fig. 1B). The overall percentage of CAR-positive cells, consisting of myc single-positive and myc/VSV-G double-positive cells, correlated well with that of the stCAR T cells (Fig. 1C). To confirm that the VSV-G-positive, vector-bound cells converted into CAR-expressing T cells, stCAR T cells were cultivated further for up to 7 days. With ongoing cultivation, the number of VSV-G-positive cells decreased, and most vector-bound cells converted into CAR-expressing cells with similar overall levels of close to 60% for both types of CAR T cells (Fig. 1D,E). Further analysis at the end-of-production time point revealed similar $CD3^+$ levels and a slightly reduced viability for stCAR T cells (Appendix Fig. S1A,B). In addition, stCAR T cells showed a less differentiated phenotype with predominantly naive T cells (50%), whereas CAR T cells were mostly central memory T cells (84%) (Fig. 1F). Moreover, exhaustion markers were generally less expressed on stCAR T cells with significantly reduced LAG-3 and TIM-3 expression (Fig. 1G). The most prominent difference was the virtual absence of LAG-3, TIM-3, and PD1 triple-positive cells in stCAR T cells compared to about 25% in conventional CAR T cells (Fig. 1H), while the cytotoxic activities were in a similar range with similar

numbers of CAR-positive cells after 24-hours killing assay (Fig. 1I; Appendix Fig. S1C). Thus, stCAR T cells were composed mostly of vector particle-bound cells with a less differentiated and less exhausted phenotype.

## In vivo evaluation of stCAR T cells

To evaluate if stCAR T cells have the potential to induce systemic adverse events, a CRS mouse model was set up. Immunodeficient NSG-SGM3 mice were engrafted with luciferase-positive NALM6 tumor cells for 10 days before stCAR T cells were injected (Fig. 2A). Tumor growth was carefully monitored by in vivo bioluminescence imaging (IVIS), showing that tumor cells were located mainly in bones of the legs, hips, and sternum (Appendix Fig. S2A–C). One day prior to stCAR T-cell injection, mice were distributed into the experimental groups based on equal tumor load (Appendix Fig. S2B). Afterward, either $1 \times 10^7$ stCAR T cells, or T cells activated and cultured under identical conditions were administered intravenously (i.v). The next morning, mice treated with stCAR T cells were in poor health, with ruffled fur, squinted eyes, and reduced responsiveness. All mice treated with stCAR T cells had to be sacrificed due to high body scores, including visual appearance, cage activity, and weight loss, whereas all T-cell-treated mice showed the same unremarkable scores as PBS-treated mice (Fig. EV1A). Only after another 10 days, the scoring index of T-cell-treated mice increased due to the tumor burden (Fig. EV1A). Strikingly, stCAR T-cell-treated mice displayed a sharp temperature drop of over 2 °C from baseline, as well as a clear drop in weight of 8.5% within 24 h (Figs. 2B and EV1B,C). Interestingly, human T lymphocytes readily detectable in peripheral blood of the control group (3% of all viable cells corresponding to 35 cells per μL blood) were absent in blood of stCAR T-cell-treated mice (Fig. 2C,D). Presumably, these had migrated to the bone marrow, where NALM6 cells (4% of viable cells), as well as infiltrated human T cells were detected with a frequency of 0.05% of all viable cells (Appendix Fig. S2D,E).

To verify that CRS caused the severe symptoms in stCAR T-cell-treated mice, human cytokines were assessed in the plasma of the mice on the day of termination and compared to blood of T-cell-treated mice obtained on the same day. The results showed a high and significant increase in various human cytokines related to CRS in all mice treated with stCAR T cells. In particular, IFN-γ was increased more than 130-fold in the stCAR T-cell group reaching 70 ng/mL in plasma. Similarly, other pro-inflammatory cytokines, such as TNF-α (114-fold), and IL-2 (6-fold), as well as the anti-inflammatory cytokine IL-10 (54-fold) were substantially increased (Fig. 2E). Interestingly, among monocyte-associated cytokines, IL-1β, was not notably elevated, whereas, IL-6, a major CRS-relevant cytokine, was significantly increased (Suppl. Fig. EV1D). Due to the multicellular interplay of the entire immune system leading to systemic adverse events in the course of CRS, murine immune cells, in particular, myeloid cells, including monocytes, macrophages, DCs, and neutrophils, all of which were detected in the spleen of NSG-SGM3 mice might have contributed to the CRS (Fig. 2F). Interestingly, the majority of murine splenocytes were neutrophils in stCAR T-cell-treated mice, whereas T-cell-treated mice showed a more prominent presence of myeloid DCs on the final day (Fig. 2F). To examine if these cells had contributed to the ongoing CRS, multiple murine cytokines were assessed in the plasma of stCAR

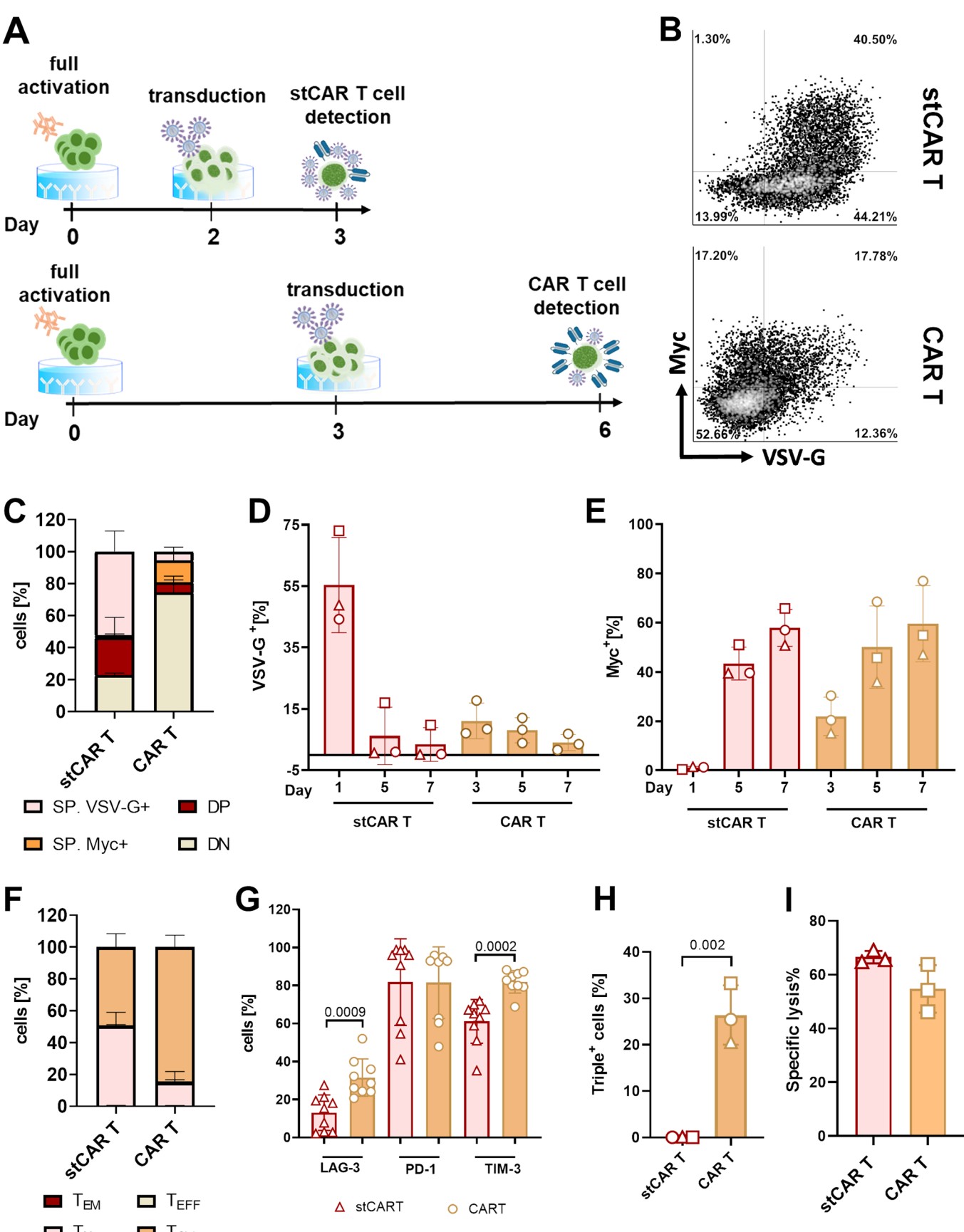

**Figure 1.  Characterization of short-term CAR T cells.**

Short-term (st) CAR T cells were characterized for the presence of vector particle components and CAR expression as well as for phenotype and exhaustion markers using flow cytometry. **(A)** The schematic illustration represents a timeline for comparing stCAR T-cell versus CAR T-cell production. **(B)** Representative dot plots for vector and CAR signals on stCAR T cells (day 3; top diagram) and CAR T cells (day 6; bottom diagram) with indicated cell frequencies inside the quadrants. Gates were set according to the non-transduced negative control. **(C)** Stacked bar diagrams providing the frequencies of single VSV-G-positive (SP VSV-G), single CAR-positive (SP CAR), double-positive (DP) and double-negative (DN) cells present in stCAR T-cell (day 3) and CAR T-cell products (day 6). **(D)** VSV-G and **(E)** myc (CAR) signals over time during the cultivation of stCAR T cells and CAR T cells. **(F)** Phenotypes, determined by CD62L and CD45RA expression, **(G)** individual exhaustion marker expression and **(H)** triple exhaustion marker expression in stCAR T cells and conventional CAR T cells. **(I)** To determine the target cell-specific lysis of NALM6 cells, an overnight killing assay at a ratio of 1:1 of effector cells to target cells was performed. Cytotoxic activity of effector cells is measured as the percent of dead CD19$^+$ cells. Data information: Mean and standard deviations are presented for average values of two replicas determined for three donors (**C–F, H, I**) or nine donors (**G**). Statistics were determined by an unpaired $t$ test with significant p values indicated. Source data are available online for this figure.

T-cell- and T-cell-treated mice on the termination day. Especially murine MCP-1 and G-CSF were significantly elevated in stCAR T cell-treated mice with an about nine-fold and 59-fold increase compared to the other group. Interestingly, murine IL-6 was slightly higher in CAR T cell than T-cell-treated mice (Figs. 2G and EV1E). Particularly MCP-1 and G-CSF, as well as the slightly enhanced release of IL-6 in stCAR T cell-treated mice, support a systemic CRS-induced acute morbidity.

## Monocytes contribute to CRS-relevant cytokine secretion

To investigate the potential impact of myeloid cells in boosting CRS induction, a co-culture was set up comprising stCAR T cells, NALM6 tumor cells and monocytes. To avoid alloreactivity, monocytes and PBMCs used for stCAR T-cell generation were derived from the same human donor (Fig. 3A). Tumor cell killing was unaffected by the presence of monocytes (Fig. 3B) and mediated by about 60% of CAR-positive cells (Fig. 3C). Analysis of cytokines secreted during cultivation in the presence of target cells revealed tremendously elevated levels of IL-6, IFN-γ, GM-CSF, and TNF-α in the supernatant of stCAR T cells cultivated with monocytes compared to T cells with or without monocytes (Fig. 3D). Of note, the presence of monocytes in the stCAR T-cell killing assay induced an about twofold upregulation of IFN-γ and IL-6 (Fig. 3D).

Next, we addressed if cytokine levels can be further boosted in NSG-SGM3 mice in the presence of human monocytes. Based on IVIS signals, tumor-engrafted NSG-SGM3 mice were distributed into six groups and treated with stCAR T, CAR T, or T cells with or without monocytes, respectively. Besides PBS-receiving mice, another control group received stCAR T cells but no tumor cells (Fig. 4A; Appendix Fig. S3). To maintain the same kinetics as in the experiment described above, we sacrificed all mice after 24 h. In this experimental setting, symptoms developed more slowly. Yet, increased body indices were observed in the majority of mice in all stCAR T-cell groups (Fig. EV2). Interestingly, the most prominent drops in weight and body temperature occurred in three mice from the group without tumors (Figs. 4B and EV2). Both types of CAR T cells were readily detectable in bone marrow with roughly twice as many conventional CAR T cells than stCAR T cells (Fig. EV3A). Further looking into the murine cell composition in the spleen, neutrophils had reached extremely high levels in mice having received stCAR T cells together with monocytes (Fig. EV3B; Appendix Table S1).

Cytokine profiling revealed a notable increase in human pro-inflammatory cytokines, including IL-2, IL-6, IL-10, TNF-α, and

IFN-γ in those mice that had received monocytes. This increase in cytokine levels was equally pronounced for stCAR T cells and conventional CAR T cells (Fig. 4C). Notably, there was no increase in cytokine levels through monocyte addition in the T-cell control group. When comparing cytokine levels between the stCAR T-cell groups and the conventional CAR T-cell groups, all mentioned cytokines, except for IFN-γ, were higher in mice that had received stCAR T cells. This was especially pronounced for IL-10 in the presence of monocytes. In line with the manifestation of symptoms, the amounts of IL-2, TNF-α, and IFN-γ in the tumor-free stCAR T-cell group were higher than in the tumor-engrafted mice and came close to those observed in the presence of monocytes (Fig. 4C).

Also, murine cytokines released from myeloid cells likely contributed to the symptoms. IL-6 and TNF-α were up in all stCAR T-cell groups compared to those having received conventional CAR T cells (Figs. 4D and EV4). This difference was especially pronounced for G-CSF. In the stCAR T group without tumor cells, G-CSF levels reached 8582 pg/ml on average, which was 69-fold higher than in the groups having received conventional CAR T cells (Fig. 4D). In contrast, plasma levels of MCP-1, CCL-4, and CXCL-1 did not differ between the stCAR T-cell and conventional CAR T-cell groups. They were, however, increased in the presence of monocytes in both groups (Figs. 4D and EV4). Thus, the presence of myeloid cells enhanced the secretion of human and murine pro-inflammatory cytokines for conventional CAR T cells and stCAR T cells.

## Only short-term CAR T cells induce CRS in the absence of target cells

As described above, typical CRS symptoms with elevated pro-inflammatory cytokines were observed in stCAR T-cell-injected mice even in the absence of tumor cells. To test if this phenomenon is unique for stCAR T cells, we compared NSG-SGM3 mice that had received either stCAR T or conventional CAR T cells (Fig. EV5A). Already four hours post-injection mild symptoms, including malaise, reduced cage activity, weight loss, and a drop in body temperature, became apparent in both groups (Fig. 5A). During the next 16 h, these symptoms changed dramatically to a severe form only in the stCAR T-cell group, resulting in termination of the experiment 26 h after applying the engineered cells (Figs. 5A and EV5B).

Human and murine cytokines were determined in blood plasma collected at the termination of the experiment. In the human profile, there was a trend towards a slight increase in the pro-

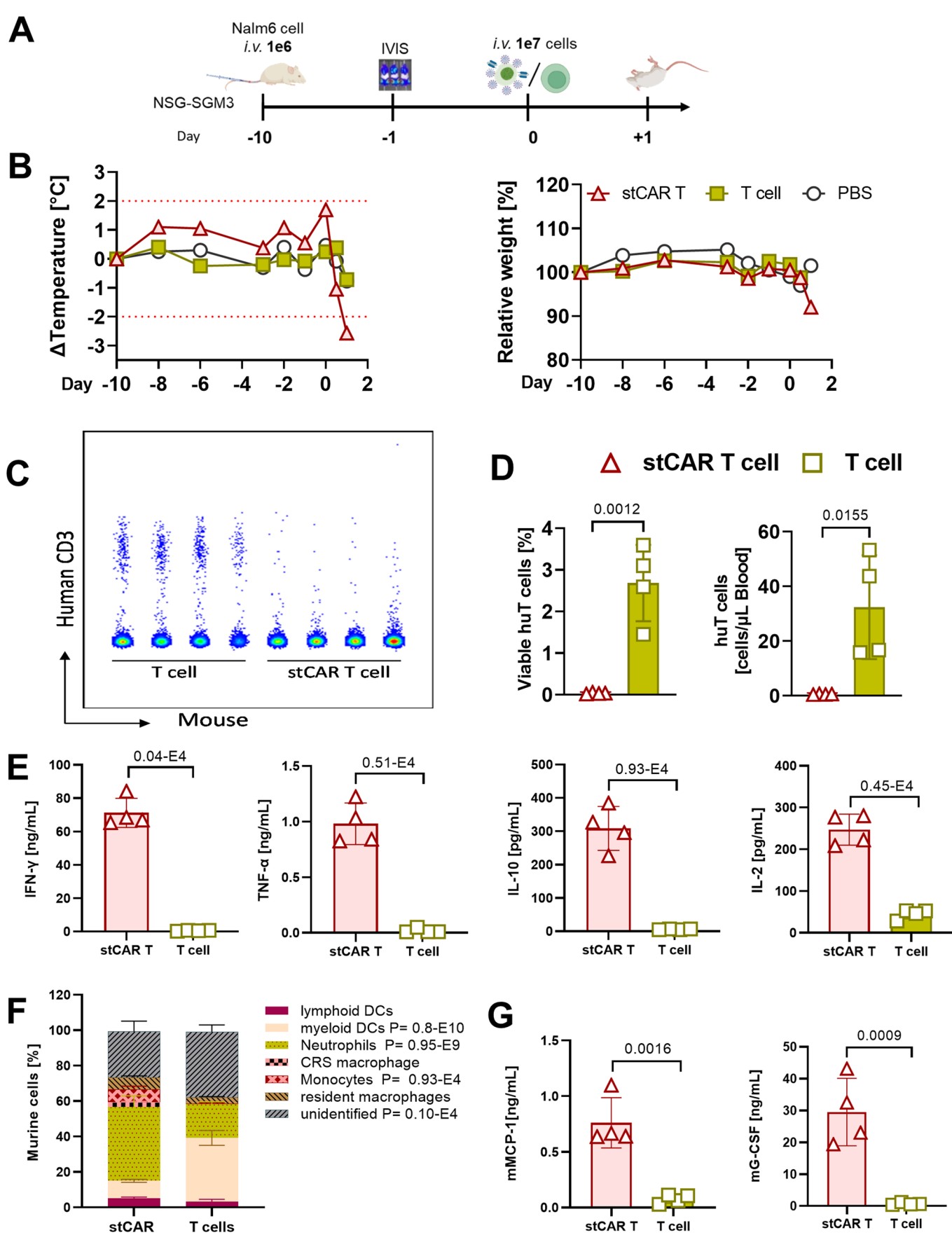

**Figure 2.    Evaluation of short-term CAR T cells in NSG-SGM3 mice.**

(A) NSG-SGM3 mice were engrafted intravenously (i.v.) with $1 \times 10^6$ EBFP- and luciferase-expressing NALM6 tumor cells. Tumor load was monitored by in vivo bioluminescence imaging system (IVIS). In all, $1 \times 10^7$ stCAR T cells or an equal amount of T cells were injected i.v. on day 0 and mice were monitored for general health conditions until termination criteria were fulfilled. (B) Body temperature change was determined as difference to experimental start at day −10. Similarly, weight was normalized to the start weight. Average values for all three groups are shown. Red dotted lines indicate the thresholds for temperature changes rated as critical adverse event. (C) Individual dot plot presentation of human $CD3^+$ T cells in blood of each mouse. (D) Bar diagrams show the frequencies of human T cells within viable cells (left) and absolute numbers per µL blood one day after stCAR T-cell or T-cell administration (right). (E) Human plasma cytokines one day after treatment. (F) Composition of murine cells in spleen. (G) Murine plasma cytokines determined on the respective termination days, which were one day after treatment for the stCAR T-cell group and 12 days after injection for the T-cell and PBS groups. Data information: Bar diagrams provide data points for each mouse with mean and standard deviation of the group. $n = 4$ for T-cell and stCAR T-cell groups, $n = 2$ for PBS group. Cytokine concentrations and body weights are the average values of two, body temperatures of three technical replicas. Statistics were assessed by unpaired $t$ test and $P$ values are indicated. Source data are available online for this figure.

inflammatory cytokines IL-6, IL-10, INF-γ, and TNF-α in the stCAR T-cell versus CAR T-cell groups, which however was not statistically significant (Fig. 5B). There was no difference in the plasma levels of IL-1β, while TGF-β was increased in the conventional CAR T-cell group (Fig. EV5C). The most pronounced difference was again seen in murine G-CSF, which was up to 45-fold higher in the stCAR T group (Fig. 5C). All other murine cytokines, including IL-6, CXCL-1, MCP-1. CCL-3, CXCL-5, and IL-1α showed no significant differences between either CAR group while being clearly increased over the PBS control (Figs. 5C and EV5D).

Assessment of lymphoid organ cell compositions further revealed a significant decline of murine neutrophils in the bone marrow of stCAR T-cell mice with an about three-fold reduction compared to control mice and a twofold reduction compared to the CAR T group (Fig. 5D). The analysis of blood showed a slight tendency for reduced neutrophil levels in the stCAR T-cell group, while within the spleen, neutrophil levels were significantly increased (Figs. 5D and  EV5E). Mouse macrophage levels in contrast were similar between both CAR groups, but when compared to the control, were elevated in bone marrow and reduced in blood (Fig. EV5F). Finally, tracking the injected human CAR T cells revealed twofold higher $CD45^+$ cell levels in bone marrow of the CAR T-cell group than the stCAR T mice (Fig. EV5G) being well in line with the relative levels of the two CAR T-cell types in the mouse studies described above.

Having seen that CRS-related cytokines were released in the absence of tumor target cells, we next asked if this can be recapitulated in the in vitro assay. Beyond that, the assay was designed such that besides stCAR T cells, T cells incubated with GFP-encoding LV or empty virus-like particles (VLPs) under the same conditions as used for stCAR T generation were included (Fig. 6A). Quantification of cytokines after 24 h gave a clear-cut picture: Neither GFP-encoding LVs nor VLPs induced cytokine release (Fig. 6B). In contrast, stCAR T cells generated with the CAR encoding LV resulted in high levels of the evaluated CRS-related cytokines (Fig. 6B). IL-2 was detectable only in presence, but not in absence, of tumor cells, while TNFα was unchanged and all other cytokines (IL-6, IFN-γ, GM-CSF, IL-10) were slightly reduced without tumor cells (Fig. 6B). Thus, also in the in vitro assay tumor cells are not required for cytokine release, whereas CAR activity is essential.

## Discussion

Current approaches to refine CAR T-cell therapy focus on the reduction of manufacturing times, from weeks down to days or

even hours. Due to the nature of CAR T cells being manufactured from patient-derived T lymphocytes, such stCAR T cells have to be considered as novel type of medicinal product potentially harboring specific safety risks. Accordingly, stCAR T cells require independent testing for activity and safety. Here, we established and evaluated assay systems for this task. The CD19-directed stCAR T cells resembled a less differentiated and exhausted phenotype when compared to conventional CAR T cells as previously described (Ghassemi et al, 2018; Zhang et al, 2022; Yang et al, 2022). Remarkably, our assay for the detection of vector components revealed that stCAR T cells consisted mainly of vector particle-bound cells with a small fraction of cells being positive for the CAR. Only after further cultivation and activation of the cells, more CAR is expressed on the cell surface, while vector particles disappear, demonstrating that transduction is not fully completed in stCAR T cells. Previous studies have not tracked the presence of vector particle components, whereas CAR transferred as a protein from packaging cells to transduced cells has been detected and studied (Ghassemi et al, 2022; Cordes et al, 2021; Jamali et al, 2019). However, CAR protein transfer alone could not mediate killing (Ghassemi et al, 2022). Using an anti-VSV-G antibody, our approach enabled accurate characterization of stCAR T cells for residual vector particles. A rather obvious safety concern that arises from the administration of large amounts of LV-bound cells is gene delivery to off-target cells. Detachment of cell surface-bound vector particles from stCAR T cells or even binding of stCAR T cells to other cells, might result in undesired gene transfer mediated by VSV-G protein. If this occurs with patient tumor cells, fatal consequences may result (Ruella et al, 2018). Accordingly, careful quality control of stCAR T cells must be ensured for a safe product.

An unexpected but highly relevant safety concern identified in our study was the fast induction of typical CRS symptoms within only 24 h after stCAR T-cell administration. This included strong ruffled fur, squinted eyes, reduced responsiveness, weight and temperature drop, as well as high levels of plasma cytokines. The pattern of increased human plasma cytokines (IFN-γ, TNF-α, IL-2, and IL-10) is in line with CRS observed in patients and notably also with the data from the in vitro cytotoxicity assay. In this assay, co-cultures of stCAR T cells and monocytes released CRS-related cytokines only when LV particles encoded the CAR, while empty or reporter-encoding LVs were ineffective. We can thus conclude that it is the interaction between CAR and residual vector components mainly contributing to the cytokine release which is further enhanced by monocytes.

Also, in vivo, after additional transplantation of human monocytes into NSG-SGM3 mice the release of CRS-relevant cytokines increased. Besides human cytokines, elevated murine cytokines, including IL-6,

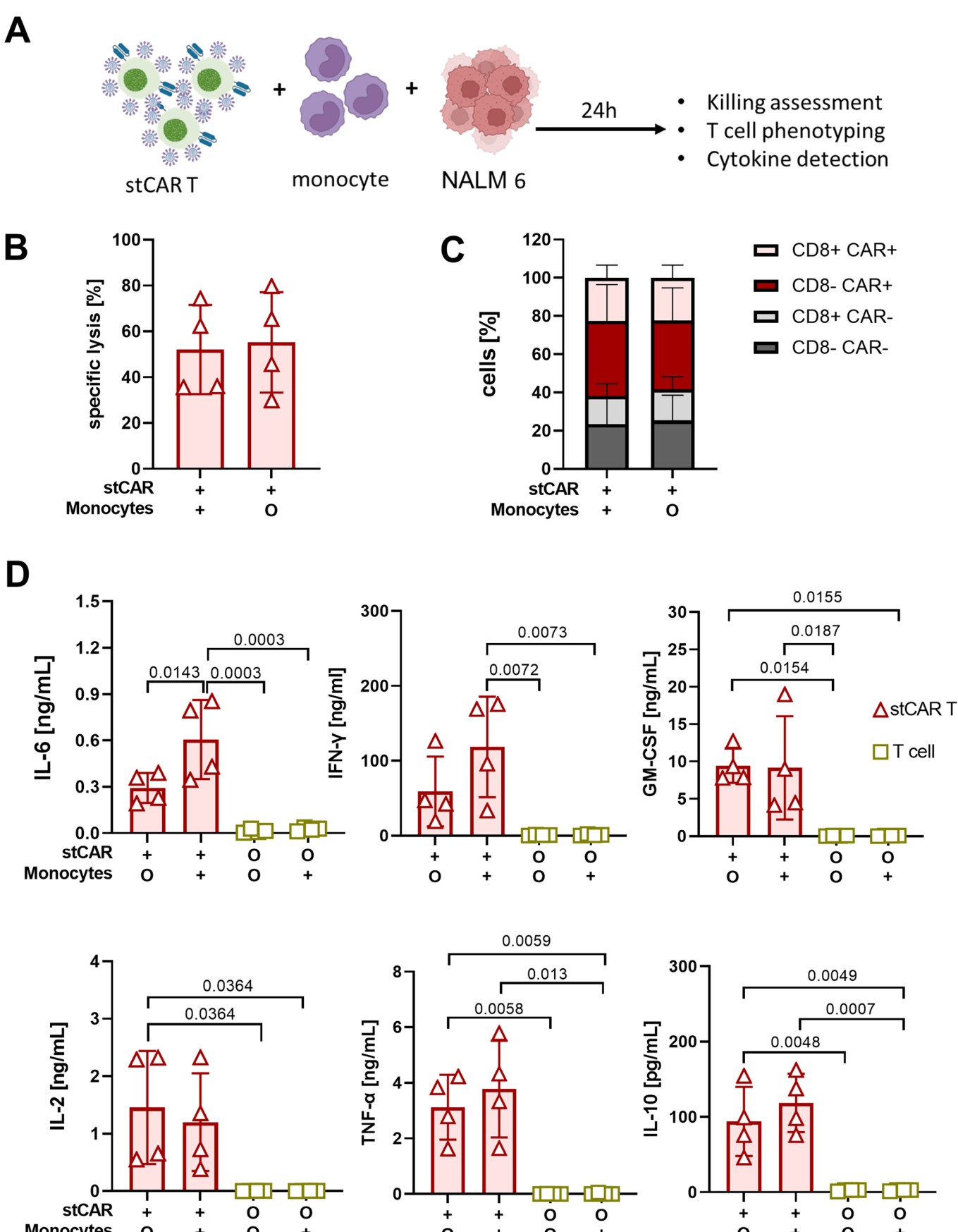

**Figure 3.   Monocytes contribute to cytokine secretion.**

(A) stCAR T cells were co-cultured with CD19$^+$ NALM6 cells in the presence (+) or absence (O) of monocytes at a 1:1:0.1 (effector: target: monocyte) ratio for 24 h. (B) Bar diagrams show the specific lysis of tumor cells determined by the amount of lysed labeled tumor cells. (C) Cell compositions of stCAR T cells after co-culture in the presence and absence of monocytes. (D) Co-culture supernatant was analyzed for human cytokines with a bead-based multiplex kit. Data information: Data are shown with standard deviations for four donors with average values of three (B, C) and two technical replicas (D). Statistics were determined by one-way ANOVA and Tukey's multiple comparisons with indicated significant P values to compare corresponding settings. Source data are available online for this figure.

G-CSF and MCP-1, underlined the ongoing multicellular interplay of human and murine cells in these mice during CRS. Immunophenotyping of residual murine immune cells in splenocytes of NSG-SGM3 mice revealed the presence of macrophages and monocytes, which presumably released murine IL-6. Furthermore, murine MCP-1, a chemoattractant for monocytes, and especially murine G-CSF, an activator for granulocytes, were particularly increased in CRS-affected mice. Both these cytokines are secreted by activated endothelial cells (Demetri and Griffin, 1991; Deshmane et al, 2009), which were described for CD19-CAR T-cell-treated patients with severe CRS and ICANS (Gust et al, 2017). In addition, G-CSF is known to induce neutrophil production and release from the bone marrow into the periphery (Ulich et al, 1988; Semerad et al, 2002) thus explaining the pronounced presence of murine neutrophils we observed in spleens of stCAR T-cell-treated mice.

Surprisingly, the CRS symptoms were not only exclusive for stCAR T cells, while absent in mice injected with conventional CAR T cells, but also independent from tumor cell injection. In this setting, the release of huge amounts of G-CSF combined with neutrophils migrating from bone marrow to spleen clearly distinguished mice having received stCAR T cells from those injected with conventional CAR T. Interestingly, G-CSF levels and kinetics are very similar in mice injected with VSV-LV (Soldi et al, 2020). It is highly remarkable, that this pattern as well as its rapid kinetic are characteristic for neutrophil reactions against viral infections (Naumenko et al, 2018). Particularly in mice intravenously injected with VSV, more than 50% of neutrophils leave the bone marrow within 24 h (Stegelmeier et al, 2020). This suggests that the high amounts of VSV-G protein on stCAR T cells trigger the innate immune response similar to a viral infection. In oncolytic viral cancer therapy this is indeed regarded as beneficial. For stCAR T cells, this remains to be seen, when more experience in clinical application has become available. The only data from clinical trials with rapidly manufactured CD19-CAR T cells documented the onset of CRS approximately 1-week post treatment (Zhang et al, 2022; Yang et al, 2022). Although this clinical study administered a rather low CAR T-cell dose ($10^4$–$10^5$ cells/kg body weight), it induced severe CRS in some patients. Our findings rather call for cautious clinical testing of this approach together with a careful determination of residual VSV-G protein on the manufactured CAR T cells.

In addition, preclinical testing and models reliably predicting safety are important. Current preclinical data about CAR T cells with shortened manufacturing time tested in therapeutic mouse models did not reveal any concerns regarding CRS and adverse events (Ghassemi et al, 2018; Zhang et al, 2022; Yang et al, 2022; Ghassemi et al, 2022). While CAR T-cell generation followed a similar protocol, the most relevant difference to our study refers to the mouse strain (NSG and NOG) used. The NSG-SGM3 mouse model used by us has previously proven to be more sensitive for CRS than the parental NSG strain when

evaluating the safety of anti-CD28 superagonist and anti-PD1 monoclonal antibodies and additionally captures the variation in cytokine release between individual donors (Ye et al, 2020). While we cannot exclude that besides the mouse strain the higher administered dose contributed to the rapid CRS induction, our results call for careful safety assessment of stCAR T cells and suggest that dose administration in the range of $10^7$ cells/kg as used with conventional CAR T cells (Maude et al, 2014; Turtle et al, 2016; Neelapu et al, 2017) may not be safe for rapidly manufactured CAR T cells. The mouse model presented here enables studies toward better understanding dosage regimens and anti-tumoral activites of stCAR T cells, as well as potential treatment strategies against CRS.

## Methods

### Vector production

The protocol for the production and characterization of VSV-LV and empty vector stocks was adapted from the previously described protocol (Weidner et al, 2021). VSV-LV contained the transfer plasmid pSEW-mycCD19-CAR encoding a second-generation CD19-CAR with a CD28 and CD3z signaling domain under control of the SFFV promotor. In brief, vector production was achieved by multiple plasmid transfection of $2.5 \times 10^7$ β2M$^{-/-}$/CD47$^{high}$ HEK293T cells in T175 flasks. After 2 days, vector particles released into the supernatant were collected and concentrated by a 20% sucrose cushion ($4500 \times g$, 24 h at 4 °C). The supernatant was discarded, the pellet was resuspended in 60 µL PBS, and the vector stocks were stored at −80 °C. The gene transfer activity of the vector stocks was determined by transduction of $8 \times 10^4$ activated human PBMCs with serial dilutions of the vector stocks. To determine particle numbers, nanoparticle tracking analysis was performed using the NanoSight NS300 (Malvern Panalytical).

### Primary cells and cell lines

Human PBMCs were isolated by Histopaque gradient centrifugation from freshly sampled human blood, purchased from the German blood donation center (DRK-Blutspendedienst, Frankfurt am Main). Blood donations were from anonymous donors. No ethics vote was required in this case as confirmed by the Ethics Committee of Goethe-University Frankfurt. Primary cells were cultured in 4Cell® Nutri-T medium (Sartorius) supplemented with 0.4% penicillin/streptomycin, 25 U/mL human IL-7 (Miltenyi Biotec) and 50 U/mL human IL-15 (Miltenyi Biotec). Monocytes were isolated from PBMCs by anti-human CD14 (REA599, APC, Miltenyi Biotec) antibody labeling and subsequent isolation using a magnetic anti-APC-Microbead Kit (Miltenyi Biotec). NALM6 cells were cultivated in RPMI 1640 medium (Biowest) supplemented with 10% fetal bovine serum (FBS)

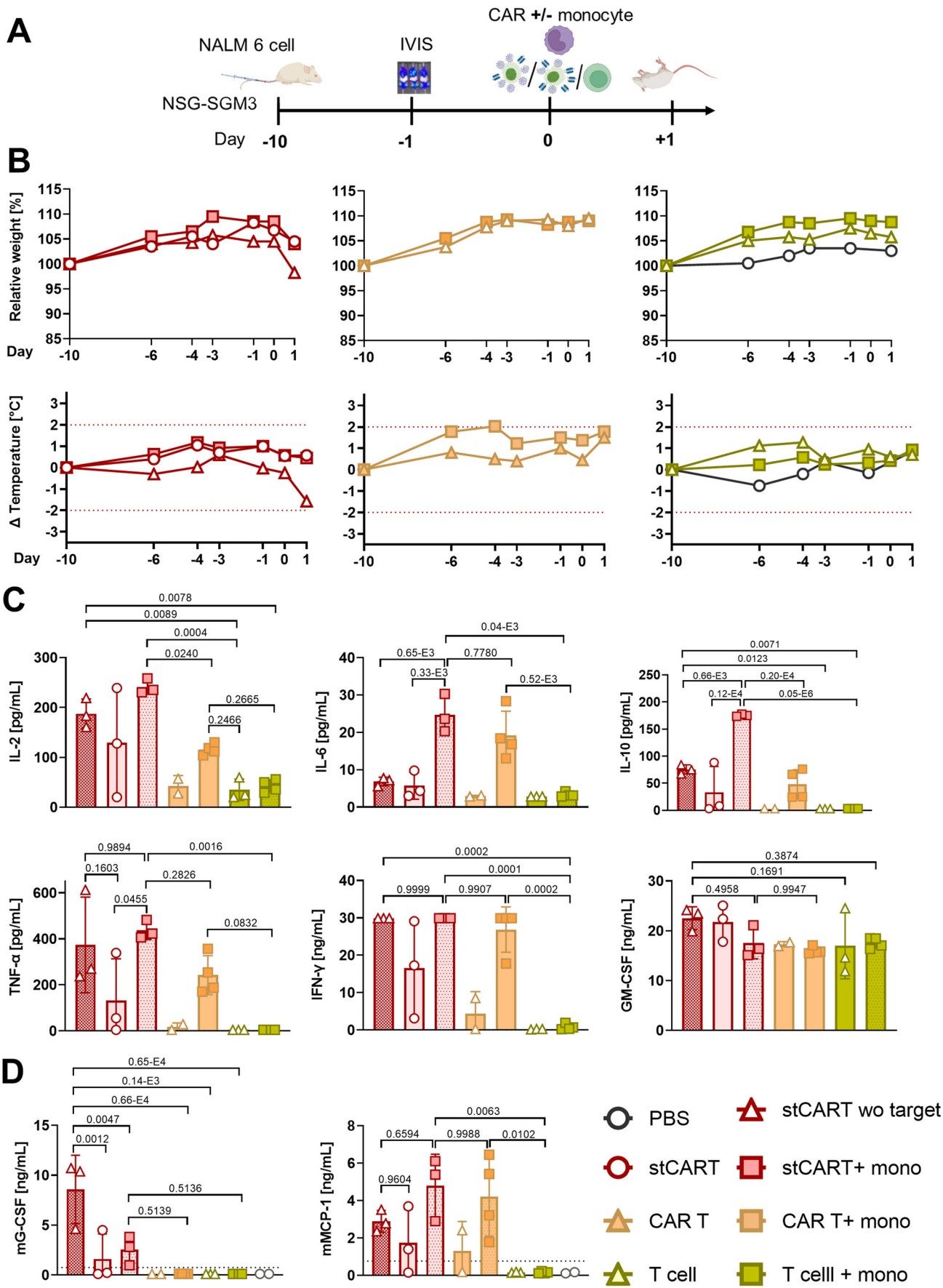

**Figure 4.  CRS induction in NSG-SGM3 mice transplanted with monocytes.**

(A) Ten days after NALM6 tumor cell injection, NSG-SGM3 mice were distributed into six groups based on equal tumor burden (Fig. EV2). Tumor-bearing mice received $1 \times 10^7$ stCAR T cellsroup, or CAR T or activated T cells (all $n = 4$), or PBS as control ($n = 2$). For each of these groups, one corresponding group ($n = 4$) supplemented with 10% human monocytes was applied. For stCAR T cells an additional group without tumor cells (wo target; $n = 4$) was implemented. Mice from all groups were sacrificed at day +1. (B) Average weights (two technical replicas) and alterations of body temperature (average values of three replicas) related to the experimental start on day $-10$. (C, D) Plasma cytokines determined for human (C) and murine (D) origin collected at experiment termination (day +1). Data information: (C, D) stCAR T cells ($n = 3$ with and without monocytes), CAR T ($n = 2$ without, $n = 3$ with monocytes), activated T cells ($n = 3$ without, $n = 4$ with monocytes), stCAR T without target cells ($n = 3$; wo target). Individual mouse data are presented as bar diagrams with mean values and standard deviation. Statistical analysis was performed by using one-way ANOVA with Tukey's multiple comparisons test. Mice with low tumor burden were excluded in statistical analysis (see Appendix. Fig. 3). Significance values are indicated. Source data are available online for this figure.

(Biochrom AG) and 2 mM glutamine (Sigma-Aldrich). $\beta 2M^{-/-}$, CD47high HEK293T cells were cultivated in DMEM (Gibco) supplemented with 10% FBS and 2 mM glutamine. All cells were cultivated at 37 °C, 5% $CO_2$, and 90% humidity. Regular testing for mycoplasma was performed on all cell lines using a PCR Mycoplasma Test Kit (PanReac Applichem, Germany).

## Conventional CAR T-cell generation

PBMCs were activated in plates coated with 1 µg/mL anti-CD3 mAb (clone: OKT3, Miltenyi Biotec) and cultured in the presence of 3 µg/mL anti-CD28 mAb (clone: 15E8, Miltenyi Biotec), 25 U/mL human IL-7 (Miltenyi Biotec) and 50 U/mL human IL-15 (Miltenyi Biotec) for 3 days. Afterward, $8 \times 10^4$ activated human PBMCs were transduced with 0.5 µL VSV-LV (MOI 4–5) via spinfection (at $850 \times g$, 90 min and 32 °C) and further cultivated for 3 days.

## Short-term CAR T-cell generation

PBMCs were activated in plates coated with 1 µg/mL anti-CD3 mAb and cultured in the presence of 3 µg/mL anti-CD28 mAb, 25 U/mL human IL-7 and 50 U/mL human IL-15 for 2 days. Next, $8 \times 10^4$ activated human PBMCs were incubated with 0.5 µL VSV-LV (MOI 4–5) in a flat-bottom 96-well plate or respectively upscaled for bigger production in a 24-well plate. Thereafter cells were cultivated for 24 h until short-term CAR T cells were harvested, washed and used for subsequent experiments. To evaluate the transition of short-term CAR T cells to fully CAR-expressing T cells, stCAR T cells were cultured in 4Cell® Nutri-T medium (Sartorius) supplemented with 25 U/mL human IL-7, and 50 U/mL human IL-15 for 3 additional days. GFP-transduced cells were produced exactly under the same conditions by replacing the packaged CAR by the GFP sequence.

## Monocyte-supplemented cytotoxicity assay

In all, $5 \times 10^4$ short-term CAR T cells (identified as VSV-G$^+$ and CAR$^+$ cells) were co-cultured with $1 \times 10^4$ NALM6 cells pre-labeled with CellTrace® Violet (CTV) (ThermoFisher) in 200 µL 4Cell® Nutri-T medium (Sartorius) supplemented with 0.4% penicillin/streptomycin in a flat-bottom 96-well plate. In addition, co-culture was further supplemented with $5 \times 10^3$ isolated autologous monocytes or without monocytes. After 24–26 h, the supernatant was taken, centrifuged at $300 \times g$ for 5 min, and transferred into a new 96-well plate, and stored at $-20$ °C until cytokine measurement. Cytotoxicity was determined as the percentage of viable cells within

CTV-positive cells and reported as the specific lysis percentage which calculated using the formula below:

$$\%Specific\ lysis = 100 \times \left( 1 - \frac{\%viable\ target\ cells\ in\ presence\ of\ CAR\ T\ cells}{\%viable\ target\ cells\ in\ presence\ of\ untransduced\ T\ cells} \right)$$

CAR T-cell level was assessed within the CTV-negative and CD3$^+$ population by flow cytometry.

## Cytokine assay

Cytokines in cell culture supernatant or plasma were analyzed with customized multiplex human or mouse cytokine Legendplex kits focusing on inflammatory cytokines (BioLegend). Samples were measured at the MACS Quant Analyzer10 (Miltenyi Biotec) and analyzed with the Legendplex software (v.8.0, BioLegend).

## CRS tumor mouse model

All animal experiments were performed in accordance with the regulations of the German animal protection law and the respective European Union guidelines. Ethic vote and animal approval was given by Regierungspräsidium Darmstadt under the identification register V 54 - 19 c 20/15 – F 107/1057. Four- to six-week-old female NSG-SGM3 (NOD.Cg-Prkdcscid Il2rgtm1Wjl Tg(CMV-IL3,CSF2,KITLG)1Eav/MloySzJ, ID: 013,062) mice were purchased from the Jackson Laboratory and housed in the animal facility of Paul-Ehrlich-Institut in accordance to standard procedures. To establish a CRS tumor model, mice were engrafted for 10 days with $1 \times 10^6$ EBFP and firefly luciferase-expressing NALM6 tumor cells by intravenous (i.v.) injection. The tumor burden of the mice was monitored twice a week using in vivo imaging (IVIS Spectrum, Perkin Elmer). For this purpose, 150 µg/g of body weight D-luciferin (Perkin Elmer) was injected intraperitoneally and imaging data were obtained 10 min later. One day prior to short-term CAR T-cell administration, mice were imaged and animals were arranged into groups for equal distribution of tumor engraftment. One day later, $1 \times 10^7$ short-term CAR T cells, defined as vector particle and CAR-positive cells, were injected i.v. As a control, the respective number of T cells, activated and cultured under the same conditions, were injected. Untreated mice, which did not receive tumor cells or T cells, were included to assess background signals in IVIS imaging and as a negative control for murine cytokine analysis. Animals were monitored for general health parameters, including visual appearance, cage activity and weight change. In addition, the temperature was measured in technical triplicates with an infrared thermometer at the anogenital

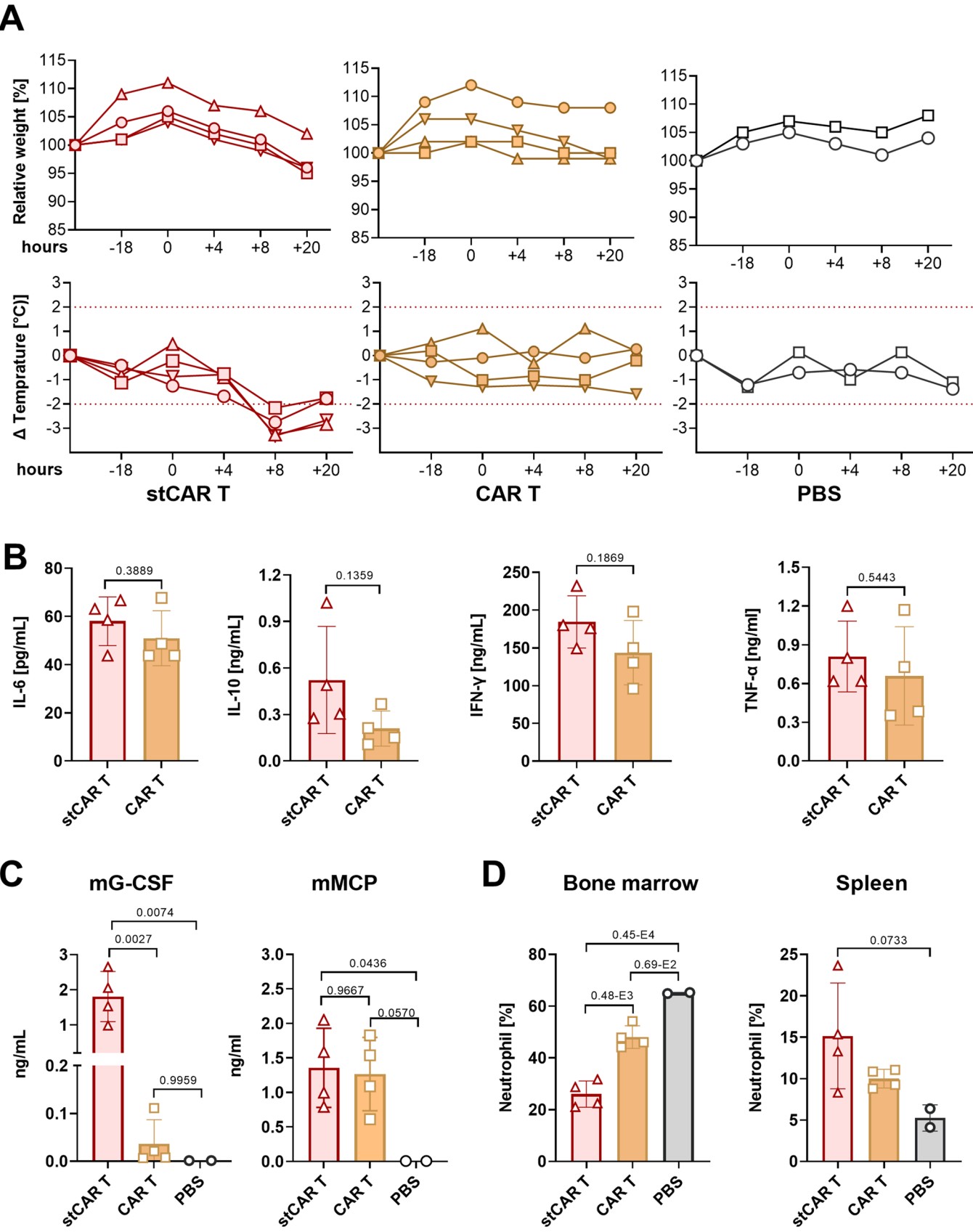

**Figure 5. Acute CRS-related adverse effects after short-term CAR T-cell administration in non-tumor-bearing mice.**

In all, $1 \times 10^7$ stCAR T cells or an equal amount of CAR T cells were injected i.v. to the NSG-SGM3 mice, at time point zero. In the following hours mice were tightly monitored for health conditions. After 24 h, the study was terminated and all mice sacrificed. (A) Weight (average values of three measurements) and body temperature (average values of two measurements) of changes in each individual mouse from 18 h before CAR T-cell injection related to the values four days before injection (CAR T-cell injection time considered as hour 0). (B, C) Plasma concentration of the indicated cytokines for human (B) and murine (C) profiles. (D) Frequencies of neutrophils in bone marrow and spleen as determined by flow cytometry analysis. Data information: $n = 4$ animals per stCAR T and CAR T groups and in PBS group $n = 2$ animals. Data are shown as mean with standard deviations and statistical significance was tested by unpaired $t$ test for (B) and one-way ANOVA with Tukey's multiple comparison test (C, D). $P$ values calculated from statistical tests are presented in corresponding graphs. Source data are available online for this figure.

area. Temperature change is presented as delta temperature, subtracted from the value prior to the experiment starting on day −10. Changes of more than 2 °C from the initial temperature value were defined as adverse events. Blood was taken 1 day after short-term CAR T-cell administration. On the final day, blood, spleen and bone marrow were harvested for further analysis.

## Preparation of single-cell suspension from organs

Blood, spleen, and bone marrow were collected from each mouse. Plasma was separated from blood cells by centrifugation at $300 \times g$ for 5 min and a second centrifugation of the supernatant at $16,000 \times g$ for 5 min to remove residual cell debris. Plasma was then stored at −80 °C until cytokine analysis. Bone marrow cells were harvested by opening ends of the bones and centrifugation at $4600 \times g$ for 3 min in perforated 0.5-mL tubes inside a fresh 1.5-mL tube containing RPMI medium. Spleens were cut into small pieces and smashed with a piston before passing through a 70-μm cell strainer. Isolated single-cell suspensions were washed with PBS and subsequently incubated in BD Pharm Lyse buffer (BD Biosciences) for 10 min for erythrocyte lysis. Isolated single-cell suspensions of the organs were used for flow cytometry analysis.

## Flow cytometry analysis

Staining for flow cytometry analysis was performed in PBS supplemented with 2% FBS. Primary cells were blocked with human FcR-blocking reagent (Miltenyi Biotec) and mouse-derived samples were additionally blocked with mouse FcR-blocking reagent (Miltenyi Biotec). The following anti-human antibodies were used for FACS staining: CD45 (2D1, BV510, BioLegend or 5B1 VioBlue Miltenyi Biotec), CD3 (BW264/56, PerCP, Miltenyi Biotec; HIT3a, BV605, BD Bioscience), CD14 (REA599, APC, or Tuk, Percp both Miltenyi Biotec), CD4 (L200,PE, BD Bioscience, VIT-4, PerCp or VioBlue, Miltenyi Biotec), CD8 (RPA-T8, BV786, BD Bioscience; BW135/80, APC or PE Miltenyi Biotec). For CAR detection anti-Myc antibody (SH1-26e7.1.3, FITC, Miltenyi Biotec) was applied. To determine cell viability, fixable viability dye (eFluor 780, ThermoFisher Scientific) was used. Stained samples were fixed in 1% PFA and stored at 4 °C until measurement. For vector particle staining against the glycoprotein of VSV, samples were incubated with the primary mouse-derived anti-VSV-G antibody (8G5F11, unlabeled, Kerafest) for 15 min at 4 °C and washed thoroughly twice. Afterward, cells were stained with a secondary anti-mouse IgG antibody (polyclonal, AF647, Jackson ImmunoResearch) for an additional 15 min at 4 °C and subsequently washed three times. Thereafter, regular antibody staining was performed as described above. Measurements were carried out on a LSR Fortessa (BD Biosciences) or MACS Quant Analyzer10 (Miltenyi Biotec), and data were analyzed using FlowJo v.10.1 (BD Biosciences)

or FCS Express 6 (De Novo Software). To determine absolute counts in the blood, defined volumes of 4 μm Count Bright Plus absolute counting beads (Invitrogen) were added to the samples before flow cytometry measurement. Phenotype of CAR T cells was determined by CD45RA (T6D11, VioBlue, Miltenyi Biotec), and CD62 (145/15, Pe-Vio770, Miltenyi Biotec) expression. Naive-like T cells were defined as CD45RA and CD62L positive, central memory T cells as CD62L single-positive, effector memory T cells as double-negative, and effector T cells as CD45RA single-positive. To evaluate the exhaustion status detected on CD3-positive/Myc positive cells using anti-LAG-3 (REA351, VioBlue, Miltenyi Biotec), anti-PD1 (PD1.3.1.3,PE-Vio770, Miltenyi Biotec) and anti-TIM-3 (7D3, PE, BD Biosciences) antibodies.

In mouse samples, tumor cells were gated on human CD45-negative cells and determined as EBFP-positive cells. Human T cells were identified as human CD3-positive and EBFP-negative cells. Murine cells present in the human CD45-negative population were further distinguished with antibodies directed against Ly-6G (REA126,PE-Vio770, Miltenyi Biotec), CD11b (M1/70, Percp, eBioscience), and CD11C (N4/18, VioBlue, eBioscience).

See Appendix Table S2 for more detailed information on all antibodies used.

## Body scoring

To assess and score the general health condition of the animals, relative weight loss, cage activity, and visual appearance were each scored separately from 0 to 3 and added together for the total body score. Healthy mice with no weight loss, normal cage activity and normal appearance were scored with 0. Animals with increasing weight loss (>0%; >5%; >10%), decreasing cage activity (moderately active, jumpy behavior; low activity, separating from others; barely responsive), and unhealthy visual appearance (slightly ruffled fur; ruffled fur with hunched back position; ruffled fur with hunched back position and squinted eyes) were scored accordingly (1; 2; 3). Mice were terminated when the total body score was higher than 4, or, a score of 4 for persisted for more than 2 consecutive days.

## Statistical analysis

Data was analyzed using the GraphPad Prism software version 8 (GraphPad Software, USA). Statistical differences were assessed as indicated in the figure legend by using unpaired $t$ test for comparison between two groups, one-way and two-way ANOVA test with Tukey's comparison test for comparing among three or more groups. Data are presented as means with standard deviation. Differences were considered significant at $P < 0.05$. For the mouse studies, sample sizes were chosen to comply with animal protection requirements and to allow statistical testing. Animals were

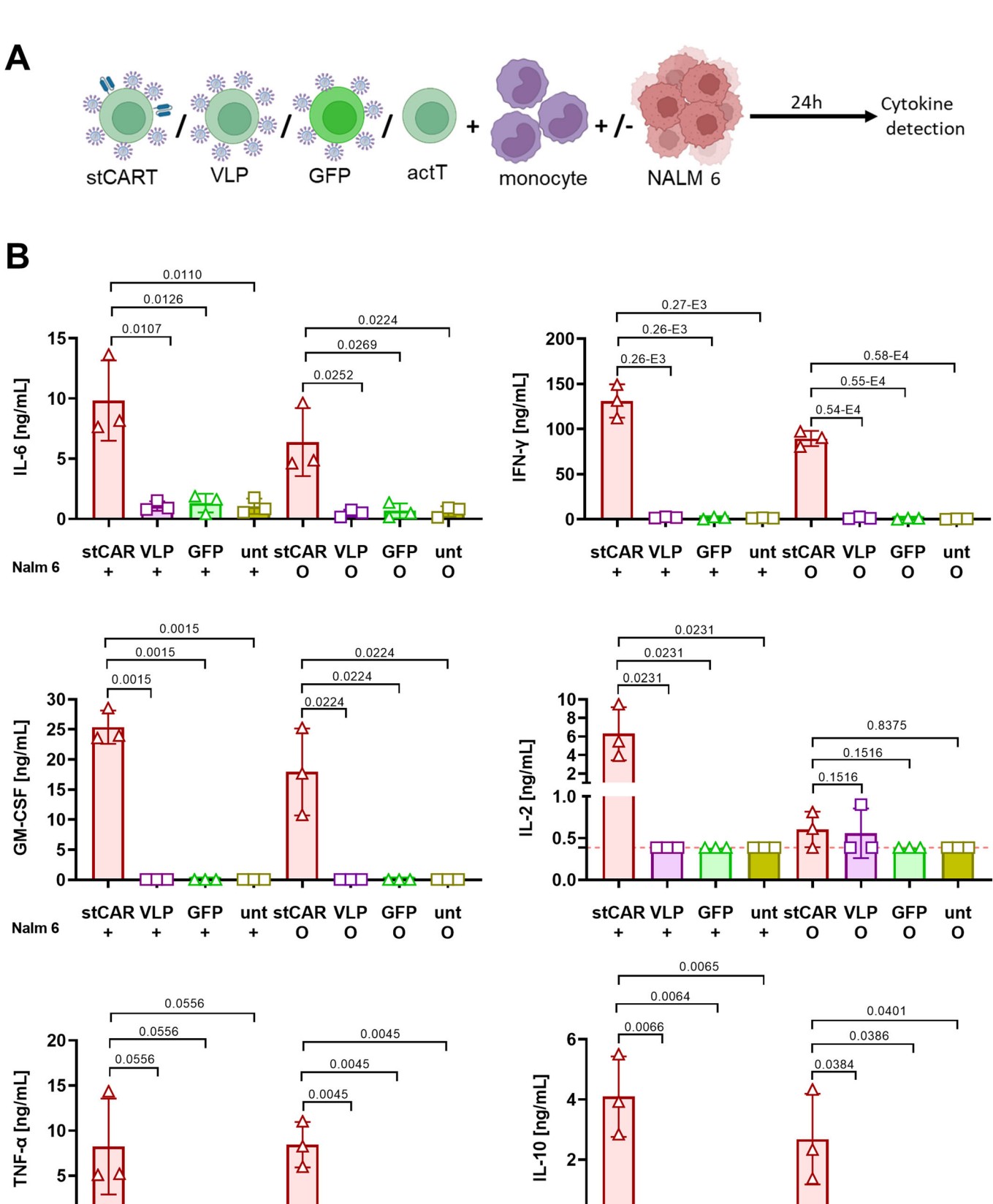

**Figure 6. The CAR but not tumor cells are causative for cytokine release.**

(A) Experimental design of the in vitro cytokine release assay. Activated T cells obtained from three different donors ($n = 3$) incubated with CAR- (stCAR T) or GFP-encoding (GFP) LVs, or empty LVs (VLP) or without particles (actT) were co-cultivated with monocytes in the presence (+) or absence (o) of NALM6 cells. (B) The indicated human cytokines were quantified 24 h after the start of the co-cultures. Data information: Mean values of two technical replicas and standard deviations are shown. Statistical significance was tested with unpaired $t$ test. $P$ values are provided. Source data are available online for this figure.

## The paper explained

### Problem

CAR T cells have brought immense benefits to patients suffering from various types of hematological malignancies. Due to their complex manufacturing not all patients in need can be treated. Among the various strategies aiming at simplifying the process, patient T cells shortly incubated with CAR-delivering vector particles have recently entered clinical testing. So far, preclinical safety testing of these short-term CAR T cells is limited due to lack of preclinical models predicting potential induction of cytokine release syndrome (CRS).

### Results

Here, we describe an easily accessible mouse model developing CRS-like symptoms within hours after injection of stCAR T cells. In addition, a cell-based assay is presented which recapitulates the release of CRS-relevant cytokines. The study shows concerningly strong CRS induction with short-term CAR T cells as compared to conventional CAR T cells. Cytokine release was independent of the presence of tumor cells and the consequence of residual vector particle components on the surface of short-term CAR T cells.

### Impact

The study calls for special attention to CRS induction after clinical use of short-term CAR T cells and provides straightforward test systems to assess the safety of newly developed short-term CAR T cells.

randomly distributed to the groups or based on equal tumor loads without blinding. Any animals that had to be removed from the groups for unexpected reasons are mentioned in the particular experimental descriptions. Detailed information on replicates and sample sizes, as well as exclusion criteria and grouping of mice, is provided in each figure legend.

## Data availability

This study includes no data deposited in external repositories.

## Peer review information

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

## Acknowledgements

The authors like to thank Gundula Braun and Manuela Gallet for excellent technical assistance. This work was supported by grants from the Bundesministerium für Gesundheit (ZMV I 1 - 25 18 FSB 404) and the Deutsche Krebshilfe (70114099) to CJB.

## Author contributions

**Arezoo Jamali**: Data curation; Formal analysis; Investigation; Visualization; Methodology; Writing—review and editing. **Naphang Ho**: Conceptualization; Data curation; Formal analysis; Investigation; Methodology; Writing—review and editing. **Angela Braun**: Formal analysis; Investigation; Visualization; Writing—review and editing. **Elham Adabi**: Investigation; Visualization; Writing —review and editing. **Frederic B Thalheimer**: Conceptualization; Supervision; Funding acquisition; Writing—review and editing. **Christian J Buchholz**: Conceptualization; Supervision; Funding acquisition; Writing—original draft; Writing—review and editing.

## Funding

## Disclosure and competing interests statement

CJB and FBT are inventors of patents describing cell-type specific lentiviral vectors. The remaining authors declare no competing interests.

# Expanded View Figures

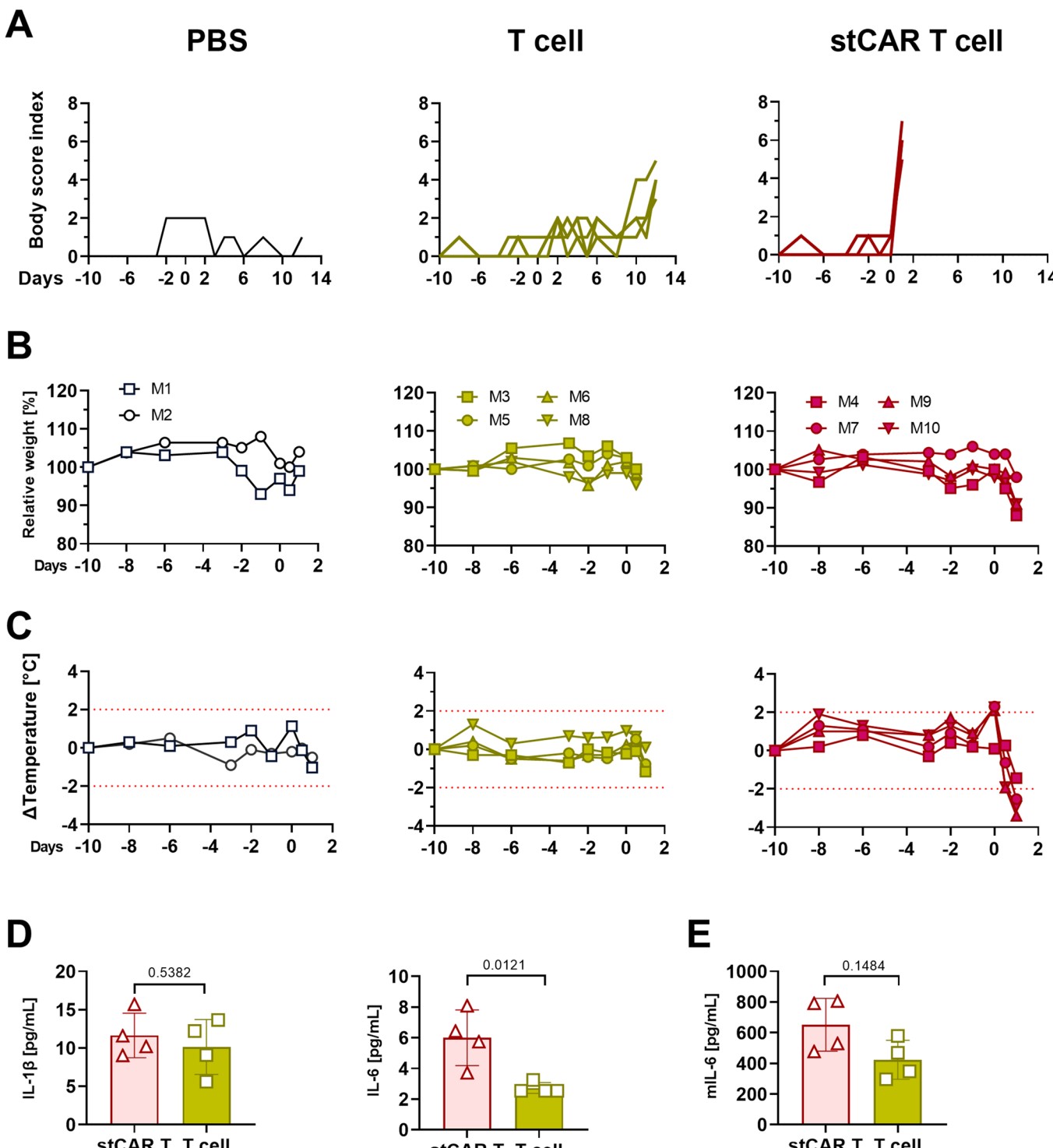

**Figure EV1.** Additional data to mouse study shown in Fig. 2.

(A) Body score indices for each mouse of the respective groups over time after cell injections. (B, C) Changes in weight and body temperature of all three groups over time. (D) Plasma levels of human IL-6 and IL-1β measured by multiplex kit after termination of the experiment. (E) Murine IL-6 level at termination day. Data information (D, E): In both groups (stCAR T cells and T cells) *n* = 4 animals. Graphs represent individual data with mean and standard deviation. Statistics were determined by unpaired *t* test with *P* values provided.

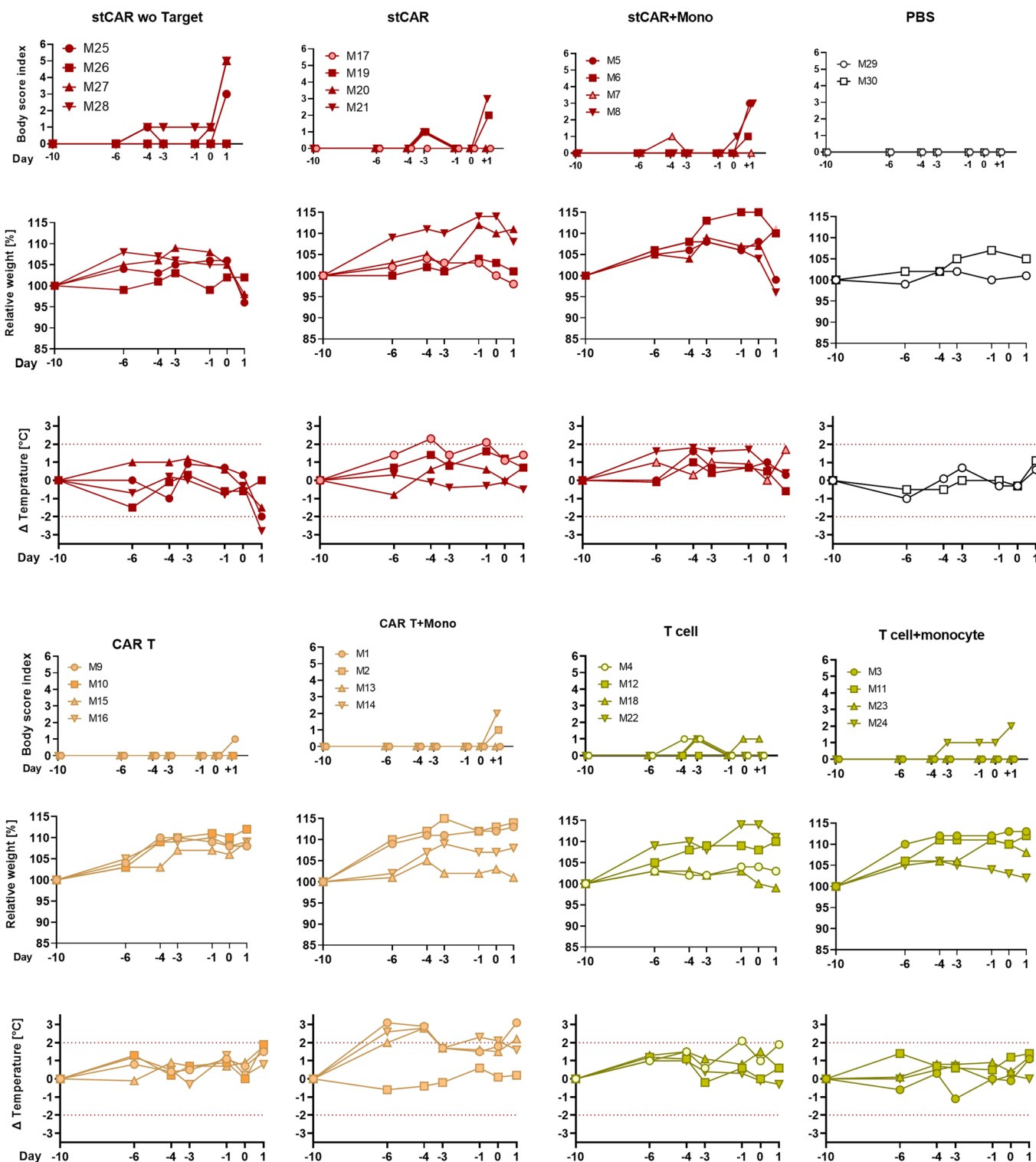

**Figure EV2. Health status of individual mice from the study shown in Fig. 4.**

Body score indices, body weights and temperatures of all mice included in the study are shown in Fig. 4. Individual animals from the groups having received stCAR T cells (red lines), PBS (black lines), conventional CAR T cells (yellow lines), and control T cells (green lines) are distinguished by the symbols provided in the top diagram, respectively. Individually calculated relative weights and changes in body temperature were related to the start time point of the study. Data for all eight groups are shown; $n = 4$ for all treated groups and $n = 2$ for PBS group.

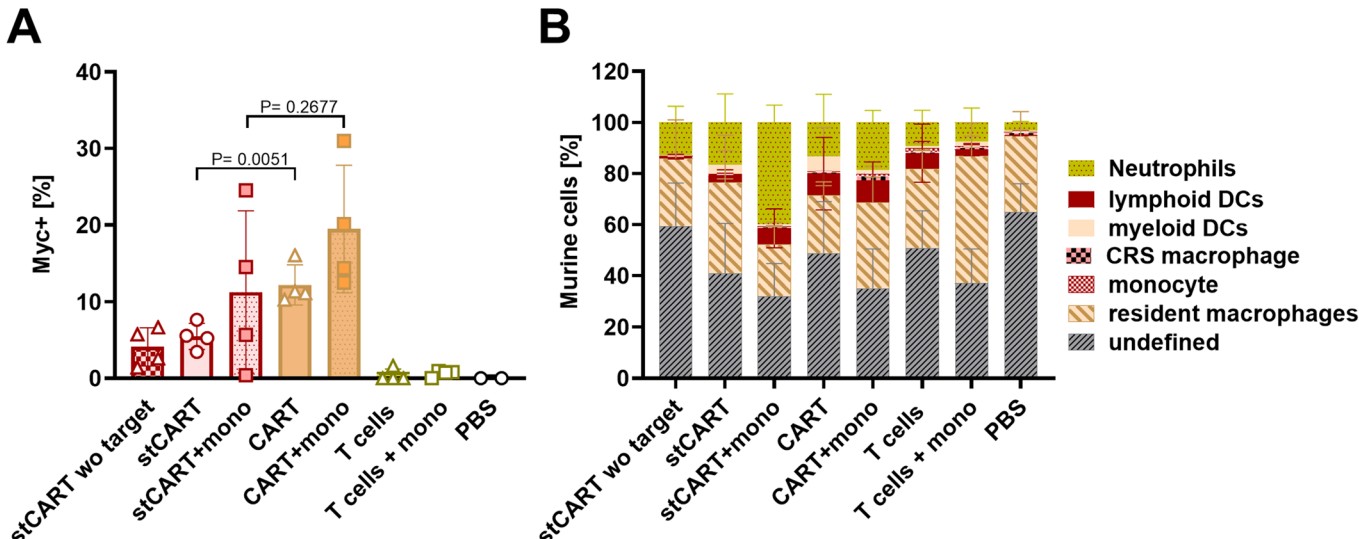

**Figure EV3.  Cellular analysis of mice from the study shown in Fig. 4.**

(A) Frequencies of CAR-positive cells in bone marrow determined by flow cytometry with antibodies against the myc tag on the CAR. Data are shown for individual mice with mean values and standard deviations for the each group. Statistical significance was tested with unpaired *t* test. *P* values are provided. (B) Composition of murine cells in the spleen from different groups shown as stacked bars with the mean and standard deviation for each fraction of the splenocyte populations. (A, B) $n = 4$ for all treated groups and $n = 2$ for PBS group. Statistical significance of pair-wise comparisons between the levels of neutrophils in all groups are shown in Appendix Table S1.

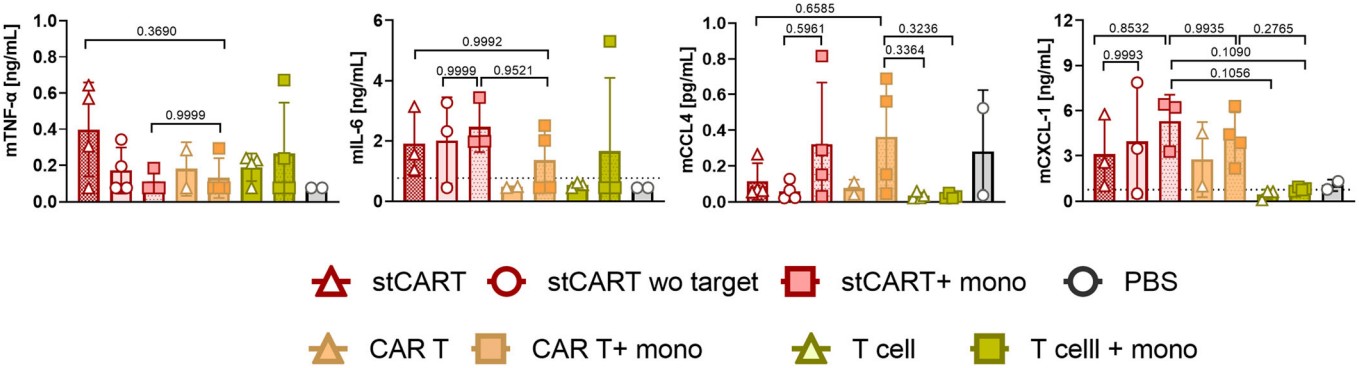

**Figure EV4. Murine cytokines in mice from the study shown in Fig. 4.**

Levels of the indicated cytokines in plasma obtained upon scarifying the animals. Values for each individual mouse are indicated. Bars represent mean values and standard deviations. Statistical analysis was performed by using one-way ANOVA with Tukey's multiple comparisons test to show mean differences among the groups. Mice with low tumor burden were excluded in statistical analysis (see Appendix. Fig. 3), which was not possible for CAR T-cell group ($n = 2$). $P$ values are provided.

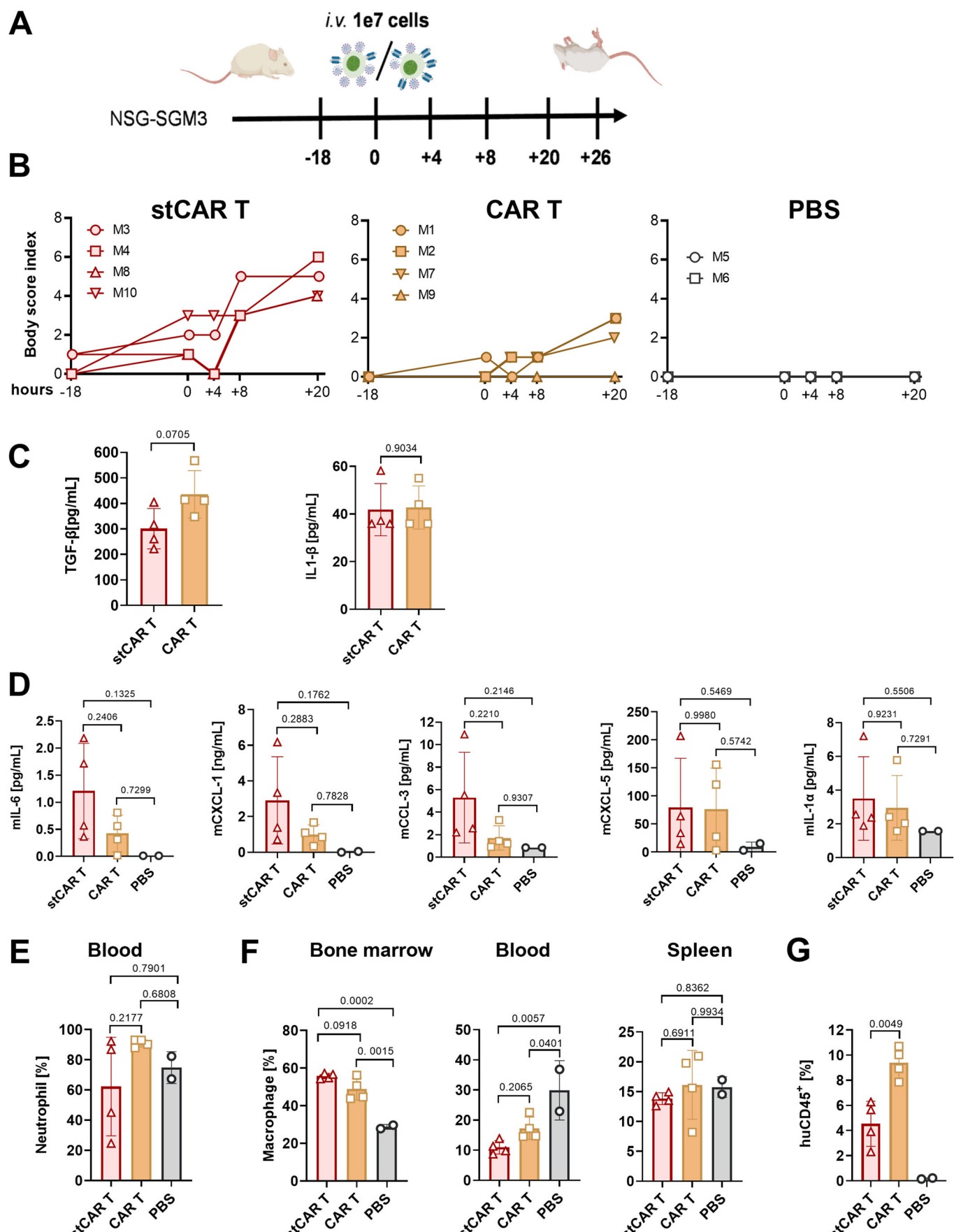

**Figure EV5. Supportive data to Fig. 5.**

(A) Experimental outline. The timeline is indicating hours. (B) Body score indices of individual animals over time. (C, D) Cytokine concentrations of human TGF-β and IL-1β (C) as well as of murine IL-6, CXCL-1, CCL-3, CXCL-5, and IL-1α (D) in plasma collected at termination time point. (E) Frequencies of murine neutrophils in blood. (F) Frequencies of murine macrophages in bone marrow, blood, and spleen. (G) Human CD45$^+$ cells in bone marrow. Data information: Bars show the mean and standard deviation for each group (C–G). Statistics were determined by unpaired $t$ test (C) two-way ANOVA with Tukey's multiple comparisons test (F). $P$ values are provided.

