## [Peer Review File · EMBO Molecular Medicine]

Early induction of cytokine release syndrome by rapidly generated CAR T cells in preclinical models

Christian Buchholz, Arezoo Jamali, Naphang Ho, Angela Braun, Elham Adabi, and Frederic Thalheimer

Corresponding author(s): Christian Buchholz (Christian.Buchholz@pei.de)

Review Timeline:

Submission Date:	27th Oct 23
Editorial Decision:	23rd Nov 23
Revision Received:	7th Feb 24
Editorial Decision:	16th Feb 24
Revision Received:	23rd Feb 24
Accepted:	4th Mar 24

Editor: Lise Roth

Transaction Report:

23rd Nov 2023

Dear Prof. Buchholz,

Thank you for the submission of your manuscript to EMBO Molecular Medicine. We have now received feedback from the three reviewers who agreed to evaluate your manuscript. As you will see from the reports below, the referees acknowledge the interest of the study and are overall supporting publication of your work pending appropriate revisions.

Addressing the reviewers' concerns in full will be necessary for further considering the manuscript in our journal, and acceptance of the manuscript will entail a second round of review. EMBO Molecular Medicine encourages a single round of revision only and therefore, acceptance or rejection of the manuscript will depend on the completeness of your responses included in the next, final version of the manuscript. For this reason, and to save you from any frustrations in the end, I would strongly advise against returning an incomplete revision.

We are expecting your revised manuscript within three months, if you anticipate any delay, please contact us.

We require:

4) A .docx formatted letter INCLUDING the reviewers' reports and your detailed point-by-point responses to their comments. As part of the EMBO Press transparent editorial process, the point-by-point response is part of the Review Process File (RPF), which will be published alongside your paper.

5) A complete author checklist, which you can download from our author guidelines (<https://www.embopress.org/page/journal/17574684/authorguide#submissionofrevisions>). Please insert information in the checklist that is also reflected in the manuscript. The completed author checklist will also be part of the RPF.

6) It is mandatory to include a 'Data Availability' section after the Materials and Methods. Before submitting your revision, primary datasets produced in this study need to be deposited in an appropriate public database, and the accession numbers and database listed under 'Data Availability'. Please remember to provide a reviewer password if the datasets are not yet public (see <https://www.embopress.org/page/journal/17574684/authorguide#dataavailability>).

7) For data quantification: please specify the name of the statistical test used to generate error bars and P values, the number (n) of independent experiments (specify technical or biological replicates) underlying each data point and the test used to calculate p-values in each figure legend. The figure legends should contain a basic description of n, P and the test applied. Graphs must include a description of the bars and the error bars (s.d., s.e.m.). Please provide exact p values.

8) Our journal encourages inclusion of *data citations in the reference list* to directly cite datasets that were re-used and obtained from public databases. Data citations in the article text are distinct from normal bibliographical citations and should directly link to the database records from which the data can be accessed. In the main text, data citations are formatted as follows: "Data ref: Smith et al, 2001" or "Data ref: NCBI Sequence Read Archive PRJNA342805, 2017". In the Reference list, data citations must be labeled with "[DATASET]". A data reference must provide the database name, accession number/identifiers and a resolvable link to the landing page from which the data can be accessed at the end of the reference.

Further instructions are available at .

9) We replaced Supplementary Information with Expanded View (EV) Figures and Tables that are collapsible/expandable online. A maximum of 5 EV Figures can be typeset. EV Figures should be cited as 'Figure EV1, Figure EV2" etc... in the text and their respective legends should be included in the main text after the legends of regular figures.

10) The paper explained: EMBO Molecular Medicine articles are accompanied by a summary of the articles to emphasize the major findings in the paper and their medical implications for the non-specialist reader. Please provide a draft summary of your article highlighting

- the medical issue you are addressing,

- the results obtained and

- their clinical impact.

11) For more information: There is space at the end of each article to list relevant web links for further consultation by our readers. Could you identify some relevant ones and provide such information as well? Some examples are patient associations, relevant databases, OMIM/proteins/genes links, author's websites, etc...

12) Author contributions: CRedit has replaced the traditional author contributions section because it offers a systematic machine readable author contributions format that allows for more effective research assessment. Please remove the Authors Contributions from the manuscript and use the free text boxes beneath each contributing author's name in our system to add specific details on the author's contribution. More information is available in our guide to authors.

13) Disclosure statement and competing interests: We updated our journal's competing interests policy in January 2022 and request authors to consider both actual and perceived competing interests. Please review the policy <https://www.embopress.org/competing-interests> and update your competing interests if necessary.

14) Every published paper now includes a 'Synopsis' to further enhance discoverability. Synopses are displayed on the journal webpage and are freely accessible to all readers. They include a short stand first (maximum of 300 characters, including space) as well as 2-5 one-sentences bullet points that summarizes the paper. Please write the bullet points to summarize the key NEW findings. They should be designed to be complementary to the abstract - i.e. not repeat the same text. We encourage inclusion of key acronyms and quantitative information (maximum of 30 words / bullet point). Please use the passive voice. Please attach these in a separate file or send them by email, we will incorporate them accordingly.

15) As part of the EMBO Publications transparent editorial process initiative (see our Editorial at <http://embomolmed.embopress.org/content/2/9/329>), EMBO Molecular Medicine will publish online a Review Process File (RPF) to accompany accepted manuscripts.

In the event of acceptance, this file will be published in conjunction with your paper and will include the anonymous referee reports, your point-by-point response and all pertinent correspondence relating to the manuscript. Let us know whether you agree with the publication of the RPF and as here, if you want to remove or not any figures from it prior to publication.

I look forward to receiving your revised manuscript.

Yours sincerely,

Lise Roth

***** Reviewer's comments *****

Referee #1 (Comments on Novelty/Model System for Author):

This is a very interesting paper exploring the murine models for assessing CRS in CAR T treatment and their data provide important clues. However, the controls used in the experiments were not ideal. This paper could improve significantly if this caveat is solved.

Referee #1 (Remarks for Author):

The authors assessed the risk of cytokine release syndrome (CRS) from short-term (st) CAR T cells in a murine model. Their data recapitulated the early onset of CRS in the murine model and emphasized the importance of assessing the stCAR T cells used in clinical trials.

Compared with standard CAR T cells, stCAR T cells showed a similar stable expression level of CARs day seven post-transduction and elicited comparable killing of target cells in vitro. However, stCAR T cells had a higher percentage of naïve T cells with a less differentiated phenotype. Infused into mice bearing NALM6 tumor, stCAR T cells triggered typical CRS-related symptoms in mice, which was probably exaggerated by monocytes. It was also very interesting that in this murine model, stCAR T cells could trigger the CRS symptoms in the absence of target tumor cells.

These findings are very interesting as there are few accepted murine models to assess the CRS, especially regarding stCAR T cells. Their data could provide helpful information for developing short-term CAR T therapies. I recommended publishing this paper with minor revisions.

Minor revisions:

1. It is very interesting that stCAR T cells alone could trigger CRS symptoms in mice without recognizing NALM6 target cells. As the authors discussed, "the high amounts of VSV-G protein on stCAR T cells trigger the innate immune response similarly to a viral infection." Therefore, it is critical to make a well-controlled experiment to elucidate this question. In Figure 2 and S2, the ideal control for the mice experiment should be mock transduced T cells using the same virus backbone in addition to untransduced activated T cells. This control could answer this fundamental question.
2. Similarly, in Figure 3, it might be possible that stCAR T cells could release CRS-related cytokines in vitro without target cells. Therefore, the ideal controls should include mock transduced T cells and effector T cells alone. The expected experiment groups should be: 1) stCAR T cell no Monocytes no NALM6, 2) stCAR T cells no Monocytes + NALM6, 3) stCAR T cells + Monocytes no NALM6, 4) stCAR T cells + Monocytes + NALM6; 5) Mock T cell no Monocytes no NALM6, 6) Mock T cells no Monocytes + NALM6, 7) Mock T cells + Monocytes no NALM6, 8) Mock T cells + Monocytes + NALM6.
3. The authors thought stCAR T cells migrate more to the bone marrow, which caused the low number of stCAR T cells in peripheral blood. It would be wonderful if the authors could run a flow with stCAR and CAR T cells using a panel of chemokine receptors and integrins. Certain chemokine receptors, such as CXCR4, have a significant role in cell migration to bone marrow.
4. In Figure S2, D and E showed the human T cells and NALM6 cell in the bone marrow. I thought the two groups were mice treated with T cells and stCAR T cells. Why was there only one histogram?
5. In Figure S9, E, there were much fewer stCAR T cells than CAR T cells in bone marrow. Could the authors discuss this regarding the data in Figure 3?
6. The calculation equation for cytotoxicity was not correctly written. It should be:
$$\% \text{ Specific lysis} = 100 \times (1 - (\% \text{ viable target cells in the presence of CAR T cells}) / (\% \text{ viable target cells in the presence of untransduced T cells}))$$

Referee #2 (Comments on Novelty/Model System for Author):

CAR-T therapy is effective but very costly and stressful for the patient. There is significant toxicity, and there is shortage of capacity. Shorter manufacture is a possible solution, but has risks which this animal study documents carefully

Referee #2 (Remarks for Author):

This animal model revealing greater toxicity due to greater cytokine levels might be used to optimise the therapy of CRS, and its prevention by optimising anticytokine therapy. This is too difficult to do in humans, but might comparing therapies to CRS in these mice be useful for patients? AntiIL6 receptor is standard of care but its never been compared to antiTNF or antiIFN γ or ant combinations.

Referee #3 (Remarks for Author):

In the manuscript entitled "Early induction of cytokine release syndrome by rapidly generated CAR T cells in a preclinical mouse model", Ho et. al. seek to observe the safety of short-term expanded CAR T (st) focusing specifically on residual vector particle expression and induction of cytokine release syndrome (CRS). Given the push of the field to use shorter expansion times for cell products, this study offers important insight for the advancement of CAR T cells in the clinic. While the study is fairly easy to follow and has great potential, there are some concerns with the presented work.

Major revisions:

1. How do CAR T compare to stCAR T in the in vitro model with Nalm6 and monocytes? Are there differences in the cytokine expression or do these look similar to the in vivo model? Given that the NSG-SGM3 strain expresses human GM-CSF, it could be hiding potential differences of this cytokine between groups. This could be explored using the in vitro model. T cells alone compared to stCAR T are not necessarily helpful to compare.
2. The authors show that CAR T are twice as prevalent in the bone marrow of non-tumor-bearing mice compared to stCAR T cells. How does their expression compare in the presence of Nalm6 tumor in NSG-SGM3 mice? This would be more relevant to treatment of cancer-bearing patients.
3. How does the murine innate cell compartment (as shown in Fig. 2F) differ between Nalm6-bearing mice treated with CAR T versus stCAR T?

Minor revisions:

1. Fig. 1C: Almost 80% of CAR T cells are double negative (DN)--is this data correct? I would expect more CAR+ by this point in culture.
2. Labeling is off in the text for Fig. 1E-I.
3. Data is not presented in order--i.e. some of Suppl. Fig. 3 is discussed in the text prior to jumping back to Supp. Fig. 2. This made it difficult to follow and would flow better if the data was introduced in the correct order.
4. In Supp. Fig. 8, changing the axis of the middle plot (CAR T) to a max of 8 to match the other two graphs will make your data more striking. As it's currently graphed, it appears that the severity of CRS is comparable between stCAR T and CAR T.
5. Interferon gamma is written as both IFN-g and INF-g in the manuscript--please choose one and make consistent.

***** Reviewer's comments *****

Referee #1 (Comments on Novelty/Model System for Author):

This is a very interesting paper exploring the murine models for assessing CRS in CAR T treatment and their data provide important clues. However, the controls used in the experiments were not ideal. This paper could improve significantly if this caveat is solved.

Thank you very much for this positive feedback. We have added further data and controls as described below.

Referee #1 (Remarks for Author):

The authors assessed the risk of cytokine release syndrome (CRS) from short-term (st) CAR T cells in a murine model. Their data recapitulated the early onset of CRS in the murine model and emphasized the importance of assessing the stCAR T cells used in clinical trials. Compared with standard CAR T cells, stCAR T cells showed a similar stable expression level of CARs day seven post-transduction and elicited comparable killing of target cells in vitro. However, stCAR T cells had a higher percentage of naïve T cells with a less differentiated phenotype. Infused into mice bearing NALM6 tumor, stCAR T cells triggered typical CRS-related symptoms in mice, which was probably exaggerated by monocytes. It was also very interesting that in this murine model, stCAR T cells could trigger the CRS symptoms in the absence of target tumor cells.

These findings are very interesting as there are few accepted murine models to assess the CRS, especially regarding stCAR T cells. Their data could provide helpful information for developing short-term CAR T therapies. I recommended publishing this paper with minor revisions.

Thank you for this positive feedback, much appreciated.

Minor revisions:

1. It is very interesting that stCAR T cells alone could trigger CRS symptoms in mice without recognizing NALM6 target cells. As the authors discussed, "the high amounts of VSV-G protein on stCAR T cells trigger the innate immune response similarly to a viral infection." Therefore, it is critical to make a well-controlled experiment to elucidate this question. In Figure 2 and S2, the ideal control for the mice experiment should be mock transduced T cells using the same virus backbone in addition to untransduced activated T cells. This control could answer this fundamental question.

We totally agree, that your proposed experimental setting will be a valuable control, however, it will not be possible to perform this additional animal experiment in due course. We therefore decided to focus on the *ex vivo* experiment you suggested below, which confirmed that the presence of CAR plus the VSV glycoprotein were causative for the cytokine release we observed. The data were added as additional main figure (Fig. 6).

2. Similarly, in Figure 3, it might be possible that stCAR T cells could release CRS-related cytokines in vitro without target cells. Therefore, the ideal controls should include mock transduced T cells and effector T cells alone. The expected experiment groups should be:

- 1) stCAR T cell no Monocytes no NALM6,
- 2) stCAR T cells no Monocytes + NALM6,
- 3) stCAR T cells + Monocytes no NALM6,
- 4) stCAR T cells + Monocytes + NALM6;
- 5) Mock T cell no Monocytes no NALM6,
- 6) Mock T cells no Monocytes + NALM6,
- 7) Mock T cells + Monocytes no NALM6,
- 8) Mock T cells + Monocytes + NALM6.

Thank you for suggesting this experimental setup, which we followed with slight adaptations. The obtained data were clear-cut in showing that release of CRS-related cytokines does not occur when the vector particles delivered GFP or were empty (VLP). In contrast, massive cytokine release was determined with stCAR T cells, which further increased in presence of monocytes. The presence of tumor cells made some difference for the secretion of particular cytokines. The data are now shown in an additional main figure (Fig. 6) and mentioned in the Discussion (line 292 and following).

3. The authors thought stCAR T cells migrate more to the bone marrow, which caused the low number of stCAR T cells in peripheral blood. It would be wonderful if the authors could run a flow with stCAR and CAR T cells using a panel of chemokine receptors and integrins. Certain chemokine receptors, such as CXCR4, have a significant role in cell migration to bone marrow.

We are sorry if we were not precise enough in our statement on the bone marrow migration. In fact, there were more CAR T cells in bone marrow after administration of conventional than of stCAR T cells. We have now added a data set quantifying CAR T cells in bone marrow of the mouse study shown in Fig. 4. Also here, there were about twice as many conventional CAR T cells in BM than stCAR T cells (Fig. EV3). The misleading statement has been rephrased (line 191-195).

4. In Figure S2, D and E showed the human T cells and NALM6 cell in the bone marrow. I thought the two groups were mice treated with T cells and stCAR T cells. Why was there only one histogram?

This shows the cells migrated to BM for the stCAR T group only, since these mice had to be sacrificed on day one after cell administration whereas the group having received T cells was sacrificed after 12 days. Therefore, we do not have values for the amounts of T cells on day one.

5. In Figure S9, E, there were much fewer stCAR T cells than CAR T cells in bone marrow. Could the authors discuss this regarding the data in Figure 3?

We are sorry for explaining this improperly. As mentioned above, we have now added the comparison stCART and conventional CART in BM for the mouse study in Fig. 4 (Fig. EV3A). The comparison related to the mouse study in Fig. 5 is now shown in Fig. EV5 (previously Fig. S9). Both data sets are in full agreement showing

about half as many stCART in BM than conventional CART. We are now better explaining this (line 191-195).

6. The calculation equation for cytotoxicity was not correctly written. It should be:
$$\% \text{ Specific lysis} = 100 \times (1 - (\% \text{ viable target cells in the presence of CAR T cells}) / (\% \text{ viable target cells in the presence of untransduced T cells}))$$

Thank you for pointing this out. The formula has been corrected.

Referee #2 (Remarks for Author):

This animal model revealing greater toxicity due to greater cytokine levels might be used to optimise the therapy of CRS, and its prevention by optimising anticytokine therapy. This is too difficult to do in humans, but might comparing therapies to CRS in these mice be useful for patients? AntiIL6 receptor is standard of care but its never been compared to antiTNF or antiIFNg or ant combinations.

Thank you very much for the positive feedback and the suggestion on using the mouse model. We do now mention this in the final sentence of the Discussion.

Referee #3 (Remarks for Author):

In the manuscript entitled "Early induction of cytokine release syndrome by rapidly generated CAR T cells in a preclinical mouse model", Ho et. al. seek to observe the safety of short-term expanded CAR T (st) focusing specifically on residual vector particle expression and induction of cytokine release syndrome (CRS). Given the push of the field to use shorter expansion times for cell products, this study offers important insight for the advancement of CAR T cells in the clinic. While the study is fairly easy to follow and has great potential, there are some concerns with the presented work.

Major revisions:

1. How do CAR T compare to stCAR T in the in vitro model with Nalm6 and monocytes? Are there differences in the cytokine expression or do these look similar to the in vivo model? Given that the NSG-SGM3 strain expresses human GM-CSF, it could be hiding potential differences of this cytokine between groups. This could be explored using the in vitro model. T cells alone compared to stCAR T are not necessarily helpful to compare.

Thank you for your positive and constructive feedback. In the revised manuscript we have added data of an additional *in vitro* experiment designed according to the suggestion by reviewer #1. The data provided in main Fig. 6 demonstrate that the cytokine release requires the presence of residual vector and the CAR. Reporter gene encoding LVs or empty LVs do not induce cytokine release. Monocytes enhance cytokine release.

2. The authors show that CAR T are twice as prevalent in the bone marrow of non-tumor-bearing mice compared to stCAR T cells. How does their expression compare in the presence

of Nalm6 tumor in NSG-SGM3 mice? This would be more relevant to treatment of cancer-bearing patients.

The requested data are now provided in Fig. EV3A.

3. How does the murine innate cell compartment (as shown in Fig. 2F) differ between Nalm6-bearing mice treated with CAR T versus stCAR T?

In Fig. EV3B we do now show quantifications of the murine immune cell populations in tumor bearing mice compared to tumor-free mice ("wo target"). There were increased levels of immune cells especially macrophages detected. We are now mentioning this on line 191-195.

Minor revisions:

1. Fig. 1C: Almost 80% of CAR T cells are double negative (DN)--is this data correct? I would expect more CAR+ by this point in culture.

The data are correct. This is a very early time point compared to a 10-21 days expansion for clinical applications. As you can see in Fig. 1E the amount of CAR positive cells is increasing over the next few days to around 60%, which is well in range of CART cells manufactured for clinical applications.

2. Labeling is off in the text for Fig. 1E-I.

Thank you for making us aware of this error. We have corrected the wrong figure references.

3. Data is not presented in order--i.e. some of Suppl. Fig. 3 is discussed in the text prior to jumping back to Suppl. Fig. 2. This made it difficult to follow and would flow better if the data was introduced in the correct order.

We have corrected this.

4. In Suppl. Fig. 8, changing the axis of the middle plot (CAR T) to a max of 8 to match the other two graphs will make your data more striking. As it's currently graphed, it appears that the severity of CRS is comparable between stCAR T and CAR T.

Thank you, we have revised as suggested (now Fig. EV5B).

5. Interferon gamma is written as both IFN-g and INF-g in the manuscript--please choose one and make consistent.

We have corrected this all over.

16th Feb 2024

Dear Prof. Buchholz,

Thank you for submitting your revised manuscript. We have now received the report from the referees who re-reviewed your manuscript. As you will see below, they are satisfied with the revisions, and I will therefore be able to accept your manuscript once the following editorial points will be addressed:

1/ Manuscript text:

- Please accept the previous changes, and only keep in track changes mode any new modification.
- We can accommodate a maximum of 5 keywords, please adjust accordingly.
- Materials and Methods:
 - o Cells: please indicate whether the cells were tested for mycoplasma contamination.
 - o Mice: please check your sentence "mice were purchased from Jackson Laboratory and". Please provide age, gender, housing and husbandry conditions of the mice.
 - o Antibodies: please provide dilutions/concentrations.
 - o Statistics: please provide a statement on sample size, exclusion/inclusion criteria, randomization, and blinding.
- Data availability: Please replace your statement by "This study includes no data deposited in external repositories."
- Author contributions: CRediT has replaced the traditional author contributions section because it offers a systematic machine readable author contributions format that allows for more effective research assessment. Please remove the Authors Contributions from the manuscript and use the free text boxes beneath each contributing author's name in our system to add specific details on the author's contribution. More information is available in our guide to authors.
- Please replace "Declaration of interests" by "Disclosure statement and competing interests". We updated our journal's competing interests policy in January 2022 and request authors to consider both actual and perceived competing interests. Please review the policy <https://www.embopress.org/competing-interests> and update your competing interests if necessary.
- References: this section should be placed before the figure legends, and references should be listed in alphabetical order.

2/ Figures and Appendix:

- Please make sure all exact p values are provided in the figures or their legends, including for ns, non-significant.
- Appendix: please add a table of content with page numbers. The nomenclature should be corrected to "Appendix Figure S1" etc, and "Appendix Table S1".
- Please address the following queries from our data editors:

Figure legends:

1. Please note that a separate 'Data Information' section is required in the legends of figures 1c-i; 2d-g; 3b-d; 5b-d; EV 1d; EV 5c-g.
2. Please note that the figure 5b does not contain any statistical parameter, kindly rectify the statistical test related information in the figure legend appropriately.
3. Please note that the statistical test related information for the legend for figure EV 5c is incorrectly labelled as 5a in the legend. This needs to be rectified.
4. Please note that in figures 3d; there is a mismatch between the annotated p values in the figure legend and the annotated p values in the figure file that should be corrected.
5. Please note that for the figures 3d, p-values and statistical tests are indicated in the legends. However, comparison for the same, "" has not been represented in the figures. Please rectify this in the figures or legends as applicable.
6. Please note that information related to n is missing in the legends of figures 4c-d; EV 3b.
7. Although 'n' is provided, please describe the nature of entity for 'n' in the legends of figures 5b-d; EV 1d-e.
8. Please note that the error bars are not defined in the legends of figures 4c-d; 5b-d; EV 1d; EV 3b; EV 5c-g.

3/ Thank you for providing Source Data. Please upload them as one file per figure, and provide the completed checklist. Source data for Figure 2G is missing.

4/ Thank you for providing The Paper Explained. Please note that this section should be included in the manuscript and should contain the following sub-sections: Problem - Results - Impact. (You may refer to previously published articles for reference).

5/ I introduced minor modifications in your synopsis text, please let me know if you agree or amend as you see fit:

To make CAR T cells available to all patients, various strategies facilitating the manufacturing process are followed. Short-term (st) CAR T cells are administered shortly after exposure to lentiviral vector (LV) particles. Here, their preclinical safety was assessed ex vivo and in vivo.

- stCAR T cells contain residual vector components on their surface, in particular the vesicular stomatitis virus (VSV) glycoprotein
- NSG-SGM3 mice develop severe CRS-like symptoms rapidly after stCAR T cell administration

- Release of CRS-typical cytokines occurs in absence of tumor cells and is enhanced by monocytes
- The interplay between CAR and LV proteins is the main trigger for cytokine release

6/ As part of the EMBO Publications transparent editorial process initiative (see our Editorial at <http://embomolmed.embopress.org/content/2/9/329>), EMBO Molecular Medicine will publish online a Review Process File (RPF) to accompany accepted manuscripts.

This file will be published in conjunction with your paper and will include the anonymous referee reports, your point-by-point response and all pertinent correspondence relating to the manuscript. Let us know whether you agree with the publication of the RPF.

I look forward to receiving your revised manuscript.

Yours sincerely,

Lise Roth

***** Reviewer's comments *****

Referee #1 (Comments on Novelty/Model System for Author):

This is a very interesting paper exploring the murine models for assessing CRS in CAR T treatment and their data provide important clues. However, the controls used in the experiments were not ideal. This paper could improve significantly if this caveat is solved.

Referee #1 (Remarks for Author):

All the questions are well addressed and I recommend publishing this paper.

Referee #3 (Remarks for Author):

Thank you to the authors for addressing my comments. I have no further concerns at this time.

All editorial and formatting issues were resolved by the authors.

4th Mar 2024

Dear Prof. Buchholz,

Thank you for sending the revised files. I am pleased to inform you that your manuscript is accepted for publication and is now being sent to our publisher to be included in the next available issue of EMBO Molecular Medicine!

If you have any questions, please do not hesitate to contact the Editorial Office.

Thank you for your contribution to EMBO Molecular Medicine.

With kind regards,

Lise Roth
